EMBO
Molecular Medicine

# Ferrochelatase is a therapeutic target for ocular neovascularization

Halesha D Basavarajappa[1,2], Rania S Sulaiman[1,3,4], Xiaoping Qi[1], Trupti Shetty[1,3], Sardar Sheik Pran Babu[1], Kamakshi L Sishtla[1], Bit Lee[5], Judith Quigley[1], Sameerah Alkhairy[1], Christian M Briggs[1], Kamna Gupta[1], Buyun Tang[2], Mehdi Shadmand[1], Maria B Grant[1,3], Michael E Boulton[1], Seung-Yong Seo[5] & Timothy W Corson[1,2,3,*]

## Abstract

Ocular neovascularization underlies major blinding eye diseases such as "wet" age-related macular degeneration (AMD). Despite the successes of treatments targeting the vascular endothelial growth factor (VEGF) pathway, resistant and refractory patient populations necessitate discovery of new therapeutic targets. Using a forward chemical genetic approach, we identified the heme synthesis enzyme ferrochelatase (FECH) as necessary for angiogenesis *in vitro* and *in vivo*. FECH is overexpressed in wet AMD eyes and murine choroidal neovascularization; siRNA knockdown of *Fech* or partial loss of enzymatic function in the *Fech*[m1Pas] mouse model reduces choroidal neovascularization. *FECH* depletion modulates endothelial nitric oxide synthase function and VEGF receptor 2 levels. FECH is inhibited by the oral antifungal drug griseofulvin, and this compound ameliorates choroidal neovascularization in mice when delivered intravitreally or orally. Thus, FECH inhibition could be used therapeutically to block ocular neovascularization.

**Keywords** age-related macular degeneration; angiogenesis; ferrochelatase; griseofulvin; heme synthesis
**Subject Categories** Neuroscience; Pharmacology & Drug Discovery; Vascular Biology & Angiogenesis

## Introduction

Ocular neovascularization is a characteristic of blinding eye diseases like retinopathy of prematurity, proliferative diabetic retinopathy, and "wet" age-related macular degeneration (AMD; Das & McGuire, 2003). These diseases cause vision loss throughout life and together are the major cause of blindness in developed countries (Das & McGuire, 2003; Penn *et al*, 2008). All are characterized by abnormal angiogenesis (neovascularization) in the posterior segment of the eye (in the retina or choroid), leading to vascular leakage that can result in hemorrhage, edema, and retinal detachment, with vision loss as a direct result. This neovascularization is driven by a number of factors, prominent among which is vascular endothelial growth factor (VEGF; Penn *et al*, 2008).

Management of neovascular eye diseases has been revolutionized by the advent of anti-VEGF biologics such as ranibizumab, bevacizumab, and aflibercept (Folk & Stone, 2010; Prasad *et al*, 2010; Hanout *et al*, 2013). However, despite these considerable advances, treatment options remain limited for a substantial fraction of patients who are non-responsive or refractory to anti-VEGF therapy (Lux *et al*, 2007); no agents acting downstream of the VEGF receptor interaction are yet approved for targeted therapy of these diseases (Abouammoh & Sharma, 2011; Lally *et al*, 2012). There is thus a pressing need to identify alternative targets in angiogenesis signaling that could form the basis for new therapeutics. To uncover novel, potentially druggable angiogenesis mediators, here we took a forward chemical genetic approach to find protein targets of an antiangiogenic natural product, cremastranone (Shim *et al*, 2004; Kim *et al*, 2007, 2008; Lee *et al*, 2014; Basavarajappa *et al*, 2015). We identify ferrochelatase (FECH) as a target of this compound and show that it is necessary for angiogenesis *in vitro* and *in vivo* and targetable with the FDA-approved small molecule drug, griseofulvin. FECH is the terminal enzyme in heme biosynthesis, responsible for catalyzing the insertion of $Fe^{2+}$ ion into protoporphyrin IX (PPIX; Hamza & Dailey, 2012). To our knowledge, we describe here for the first time FECH's role as a druggable mediator of angiogenesis.

1 Eugene and Marilyn Glick Eye Institute and Department of Ophthalmology, Indiana University School of Medicine, Indianapolis, IN, USA
2 Department of Biochemistry and Molecular Biology, Indiana University School of Medicine, Indianapolis, IN, USA
3 Department of Pharmacology and Toxicology, Indiana University School of Medicine, Indianapolis, IN, USA
4 Department of Biochemistry, Faculty of Pharmacy, Cairo University, Cairo, Egypt
5 College of Pharmacy, Gachon University, Incheon, South Korea
*Corresponding author. Tel: +1 317 274 3305; E-mail: tcorson@iu.edu

# Results

### Ferrochelatase is a target of an antiangiogenic compound

To identify potentially novel protein modulators of angiogenesis, we used photoaffinity chromatography to search for targets of the naturally occurring antiangiogenic compound, cremastranone (Fig 1A), which has selective antiproliferative effects on endothelial cells (Lee *et al*, 2014). Protein binding partners of cremastranone were pulled down from a tissue lysate using immobilized cremastranone-based affinity reagent but not a negative control reagent (Fig 1B). One of the two pulled-down proteins was identified using peptide mass fingerprinting as FECH (Fig 1C and Appendix Fig S1). Immunoblot of eluates from photoaffinity pull-down experiments confirmed the identity of the pulled-down protein using an antibody against FECH (Fig 1D). In order to confirm specificity of binding between cremastranone and pulled-down proteins, the affinity reagent was incubated with tissue proteins in the presence of excess active cremastranone isomer (Fig 1B; Basavarajappa *et al*, 2014). Under this condition, the binding of FECH to the affinity reagent was markedly decreased (~87%), indicating competition for binding to FECH between the cremastranone isomer and the affinity reagent (Fig 1E).

Recombinant FECH also interacted with the affinity reagent (Fig 1F and G), indicating that the interaction does not require eukaryotic accessory proteins. Some binding to the negative control is likely due to the abundance of recombinant protein present; recent work confirms that cellular FECH does not readily bind linkers such as our control compound (Park *et al*, 2016). Moreover, cremastranone treatment of human retinal endothelial cells (HRECs) caused a dose-dependent buildup of the FECH substrate PPIX (Fig EV1A), indicative of FECH inhibition, and addition of excess 5-aminolevulinic acid (5-ALA; the first precursor compound in the heme biosynthetic pathway that promotes increased heme production) partially rescued cremastranone's antiproliferative effects on HRECs (Fig EV1B). Cremastranone did not chelate iron, suggesting that it does not act indirectly on FECH by sequestering FECH's $Fe^{2+}$ substrate (Fig EV1C). The FECH pathway, therefore, is targeted by a known antiangiogenic compound, suggesting that this protein and pathway are important for angiogenesis.

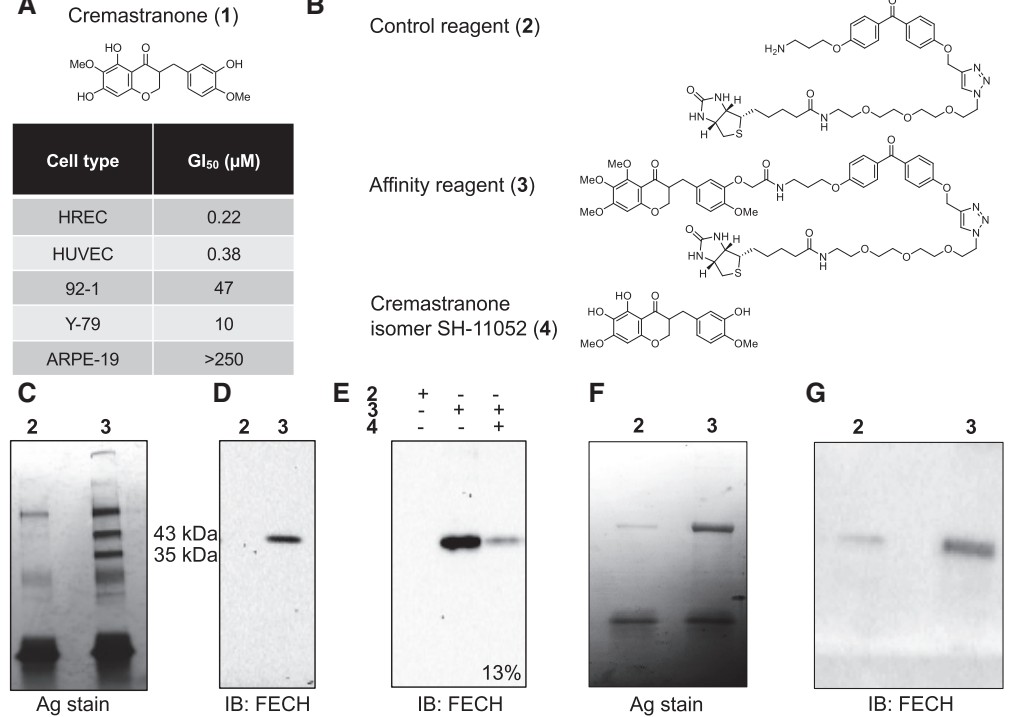

**Figure 1. Ferrochelatase (FECH) is a target of the antiangiogenic natural product, cremastranone.**

A   Chemical structure of cremastranone (**1**) (top). Antiproliferative activity of cremastranone (bottom) shown as the 50% growth inhibitory concentration ($GI_{50}$) on human retinal endothelial cells (HRECs), human umbilical vein endothelial cells (HUVECs), and non-endothelial ocular cells (uveal melanoma 92-1, retinoblastoma Y-79, and retinal pigment epithelium ARPE-19).

B   Structures of compounds used in photoaffinity chromatography.

C   Proteins pulled down with indicated reagents (numbered as in B) in photoaffinity chromatography were separated on SDS–PAGE and silver-stained.

D   Immunoblot of pulled-down proteins using antibody against ferrochelatase (FECH).

E   Immunoblot of pulled-down proteins from competition assay with excess active cremastranone isomer (**4**); relative quantification of band intensity shown.

F   Silver-stained SDS–PAGE gel of recombinant human FECH protein pulled down using photoaffinity chromatography.

G   Anti-FECH immunoblot of recombinant human FECH protein pulled down using photoaffinity chromatography.

Data information: All the gel and immunoblot images are representative from at least two independent experiments.
Source data are available online for this figure.

## Ferrochelatase is required for angiogenesis *in vitro*

The role of FECH in angiogenesis has not previously been explored, despite strong evidence linking heme catabolism with angiogenesis (Dulak *et al*, 2008). To determine whether FECH plays a key role in angiogenesis, we knocked *FECH* down in HRECs using siRNA (Fig 2A and B) and monitored key angiogenic properties of HRECs *in vitro*. *FECH* knockdown significantly reduced the proliferation of HRECs (Figs 2C and EV2A). There was also a significant decrease in migration of HRECs treated with *FECH* siRNA in a scratch-wound assay (Fig 2D). Further, knocking down *FECH* in HRECs completely abolished the tube formation ability of HRECs as monitored in a Matrigel assay (Fig 2E).

To confirm the knockdown results, we also tested whether a known pharmacological inhibitor of FECH shows antiangiogenic properties *in vitro*. *N*-methyl protoporphyrin (NMPP), a competitive inhibitor of FECH activity (Cole & Marks, 1984), also inhibited proliferation, migration, and tube formation ability of HRECs *in vitro* (Fig 2F–H). Similar results were seen in the primate choroidal endothelial cell line, Rf/6a (Fig EV3A and B), broadening the relevance of these findings. However, despite these potent antiproliferative effects, FECH knockdown and low-dose chemical inhibition were not associated with significant apoptosis of these cells (Fig EV2B–D), indicating a cytostatic rather than cytotoxic effect. This was further confirmed in a washout experiment: HRECs resumed proliferation after removal of NMPP (Fig EV2E). Moreover, *FECH* knockdown did not inhibit proliferation of non-endothelial ocular cell lines ARPE-19 and 92-1 as well as macrovascular human umbilical vein endothelial cells (HUVECs; Fig EV4A–C), indicating that FECH inhibition is not associated with general cytotoxicity. Likewise, NMPP had negligible antiproliferative effect on these cells and on retinoblastoma cell line Y-79 (Fig EV4D–F and J). However, NMPP had similar antiproliferative potency on human brain microvascular endothelial cells (BMECs) to its effects on HRECs (Fig EV4L). Together, these experiments confirm that FECH function is required for angiogenesis *in vitro* and that the angiostatic effects of FECH blockade are not cytotoxic and may be microvascular cell-specific.

## Ferrochelatase is upregulated during neovascularization

Given the potent antiangiogenic effects of *FECH* knockdown we observed in culture, we then explored whether FECH is associated with neovascularization *in vivo*. We employed a mouse model of laser-induced choroidal neovascularization (L-CNV). This widely used model recapitulates some of the features of wet AMD (Montezuma *et al*, 2009). FECH was overexpressed in and around lesions during neovascularization in this model (Fig 3A); this overexpression was seen throughout the retina, but especially in neovessels (Fig 3B). More importantly, FECH expression was seen throughout the retinal layers of human wet AMD patients analyzed postmortem. In the subretinal layers including the choroid (the origin of neovascularization in wet AMD), expression was significantly increased compared to healthy, age-matched eyes (Fig 3C).

## Ferrochelatase is required for neovascularization *in vivo*

Since FECH upregulation suggested a role for this protein in neovascularization in the living eye, we asked whether decreased FECH would inhibit this process. When L-CNV mice were treated intravitreally with *Fech*-specific siRNA, there was a significant decrease in choroidal neovascularization as compared with both saline-treated control mice and control non-targeting siRNA-treated mice (Fig 3D). To confirm these findings in a genetic model, we turned to the *Fech*^m1Pas mouse, which carries a partial loss-of-function M98K point mutation in the *Fech* gene (Tutois *et al*, 1991). In M98K heterozygotes, FECH activity is 45–65% of normal, while in homozygotes, activity is < 10% of normal (Boulechfar *et al*, 1993). We observed a reduction in L-CNV in heterozygous *Fech*^m1Pas mutant mice compared with wild type, and this reduction was even more pronounced in the *Fech*^m1Pas homozygotes (Fig 3E). These *in vivo* experiments confirm the clinical relevance of FECH in neovascularization, and the value of targeting this enzyme to block this process.

## Ferrochelatase-targeting therapy treats neovascularization

The FDA-approved antifungal drug, griseofulvin, has been in clinical use for over half a century (Petersen *et al*, 2014). The primary antifungal mechanism of this compound is as a microtubule inhibitor (Borgers, 1980). However, an unexpected off-target effect of this drug is inhibition of FECH (Holley *et al*, 1990; Brady & Lock, 1992; Martinez *et al*, 2009). Griseofulvin alkylates the heme prosthetic group of cytochrome P450 *in vivo*, forming NMPP, the FECH active-site inhibitor (Liu *et al*, 2015). Taking advantage of this property, we treated HRECs with griseofulvin and observed dose-dependent antiproliferative effects, inhibition of migration, and inhibition of tube formation (Fig 4A–C) similar to those observed after *FECH* knockdown. The concentrations of griseofulvin needed to have effects on endothelial cells were higher than those seen with NMPP (Fig 2F–H), likely due to incomplete alkylation of heme in griseofulvin-treated cells. However, this concentration (~10 μM or ~3.5 ng/ml) is 2.75 logs less than that attained in plasma during antifungal treatment of humans (1–2 μg/ml; Epstein *et al*, 1972), suggesting that efficacy *in vitro* can be achieved in a clinically attainable concentration range. Moreover, effective antiangiogenic concentrations of griseofulvin were not associated with apoptosis (Fig EV2F) of HRECs and could be washed out with resumption of proliferation (Fig EV2G). Griseofulvin did not have pronounced antiproliferative effects on other ocular cell types or HUVECs (Fig EV4G–I and K), but did affect proliferation and tube formation of Rf/6a choroidal endothelial cells (Fig EV3C and D) and proliferation of BMECs (Fig EV4M). In addition, griseofulvin inhibited formation of microvascular sprouts in the choroidal sprouting assay, an *ex vivo* model of choroidal angiogenesis (Fig 4D). It also performed similarly in an *ex vivo* retinal endothelial sprouting assay, showing pronounced antiangiogenic activity (Fig 4E). Thus, griseofulvin specifically inhibits the growth of microvascular endothelial cells, including those of the retina and choroid.

Spurred by these findings, we tested griseofulvin as a therapy for L-CNV. *Ad libitum* feeding of ≥ 0.5% (w/w) griseofulvin to mice has previously been demonstrated to induce inhibition of FECH (Brady & Lock, 1992). Excitingly, such oral treatment of L-CNV mice decreased neovascularization (Figs 5A–C and EV5A and B); similarly, dose-dependent results were seen when griseofulvin was intravitreally injected instead; this is the standard delivery route for existing anti-VEGF agents (Fig 5D–F). Both delivery routes had

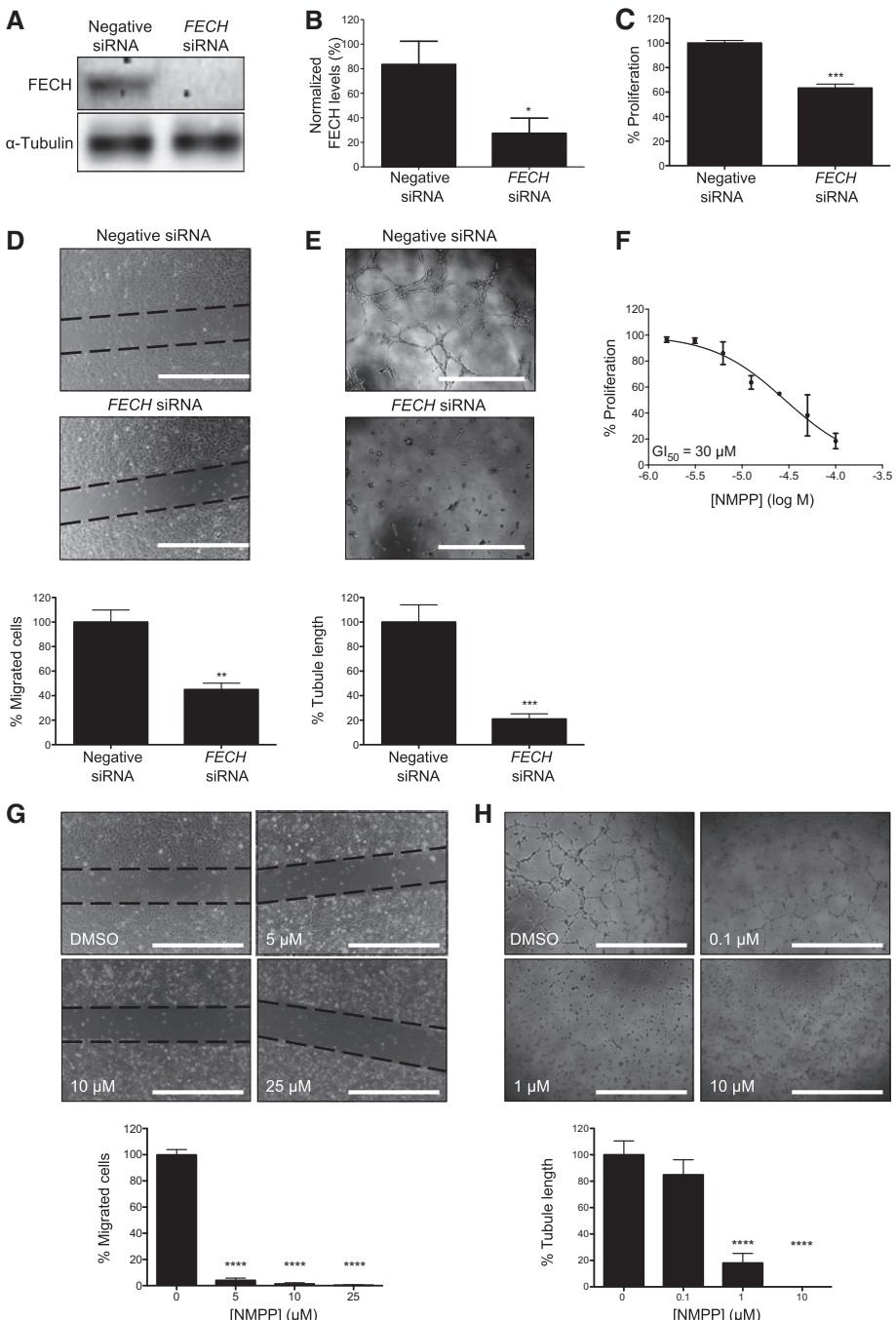

**Figure 2. FECH is an essential protein for angiogenesis *in vitro*.**

A   *FECH* is knocked down using specific siRNAs as confirmed by immunoblot.

B   Quantification of *FECH* knockdown. *P = 0.0319, two-tailed unpaired Student's *t*-test (n = 3 per group).

C   Proliferation of HRECs was monitored in presence or absence of *FECH*-specific siRNA. ***P = 0.0004, two-tailed unpaired Student's *t*-test (n = 6 per group).

D   Scratch-wound migration assay was performed with or without *FECH* knockdown in HRECs. **P = 0.0077, two-tailed unpaired Student's *t*-test (n = 3 per group). Dotted lines indicate borders of scratch at time zero.

E   Ability of HRECs to form tubes *in vitro* on Matrigel was monitored after knocking down *FECH*. ***P = 0.0003, two-tailed unpaired Student's *t*-test (n = 4 per group).

F   The effect of NMPP, a specific inhibitor of FECH activity, on *in vitro* proliferation was measured using an AlamarBlue assay (n = 3 per dose).

G   Migration of HRECs in presence of different doses of NMPP was measured using a scratch-wound assay. ****P = 0.0001 compared to DMSO-treated sample, ANOVA with Dunnett's *post hoc* tests (n = 3 per group). Dotted lines indicate borders of scratch at time zero.

H   Ability of HRECs to form tubular structures in Matrigel was monitored after NMPP treatment. ****P = 0.0001 compared to DMSO-treated sample, ANOVA with Dunnett's *post hoc* tests (n = 4 per group).

Data information: Graphs show mean ± SEM. Representative results from at least three independent experiments. Scale bars = 1 mm.
Source data are available online for this figure.

**Figure 3. FECH is an essential protein for angiogenesis *in vivo*.**

A   Whole mount staining of L-CNV mouse choroid stained with an antibody against FECH (red) and with agglutinin (green) to label neovascularization. Scale bar = 50 μm.

B   Sections of L-CNV and untreated mouse eyes stained with an antibody against FECH (red), with agglutinin (green) to label neovascularization and with DAPI to label nuclei (blue). Retinal layers indicated: GCL, ganglion cell layer; INL, inner nuclear layer; ONL, outer nuclear layer. Scale bar = 20 μm.

C   Immunostaining of sections of eyes from an 89-year-old wet AMD patient and age-matched control using an antibody against FECH (red). The nuclei of cells are stained blue with DAPI. Retinal layers indicated: GCL, ganglion cell layer; INL, inner nuclear layer; ONL, outer nuclear layer. Scale bar = 20 μm. The staining intensity of subretinal FECH, where CNV occurs, was quantified in human subjects ($n$ = 3 per group). *$P$ = 0.0269, two-tailed unpaired Student's *t*-test.

D   Whole mount staining of RPE/choroid isolated from L-CNV mice treated with *Fech*-specific siRNA. The choroidal vasculature was stained with agglutinin conjugated with Alexa Fluor 555 (red), and the neovascular area around the lesions was quantified. *$P$ = 0.044; **$P$ = 0.0038, ANOVA with Tukey's *post hoc* tests ($n$ = 5–9 per group). Scale bar = 50 μm.

E   Whole mount staining of RPE/choroid isolated from mice wild-type (WT), heterozygous (Hetero), or homozygous (Homo) for the *Fech*^m1Pas point mutation that underwent L-CNV. The choroidal vasculature was stained with agglutinin conjugated with Alexa Fluor 555 (red), and the neovascular area around the lesions was quantified. *$P$ = 0.015; **$P$ = 0.0037 versus WT, ANOVA with Tukey's *post hoc* tests ($n$ = 9–13 per group). Scale bar = 50 μm.

Data information: Graphs show mean ± SEM.

similar maximal effects. We also tested whether intravitreal griseofulvin could synergize with anti-VEGF$_{164}$ antibody treatment, equivalent to the standard of care. We observed a modest additive effect of low-dose anti-VEGF$_{164}$ with 50 μM griseofulvin (Fig EV5C), perhaps warranting further exploration of this combination therapy.

**Ferrochelatase depletion decreases the VEGF receptor via eNOS**

In order to understand how FECH contributes to angiogenesis, we knocked down *FECH* in HRECs and monitored levels of endothelial nitric oxide synthase (eNOS), a key angiogenesis regulator that

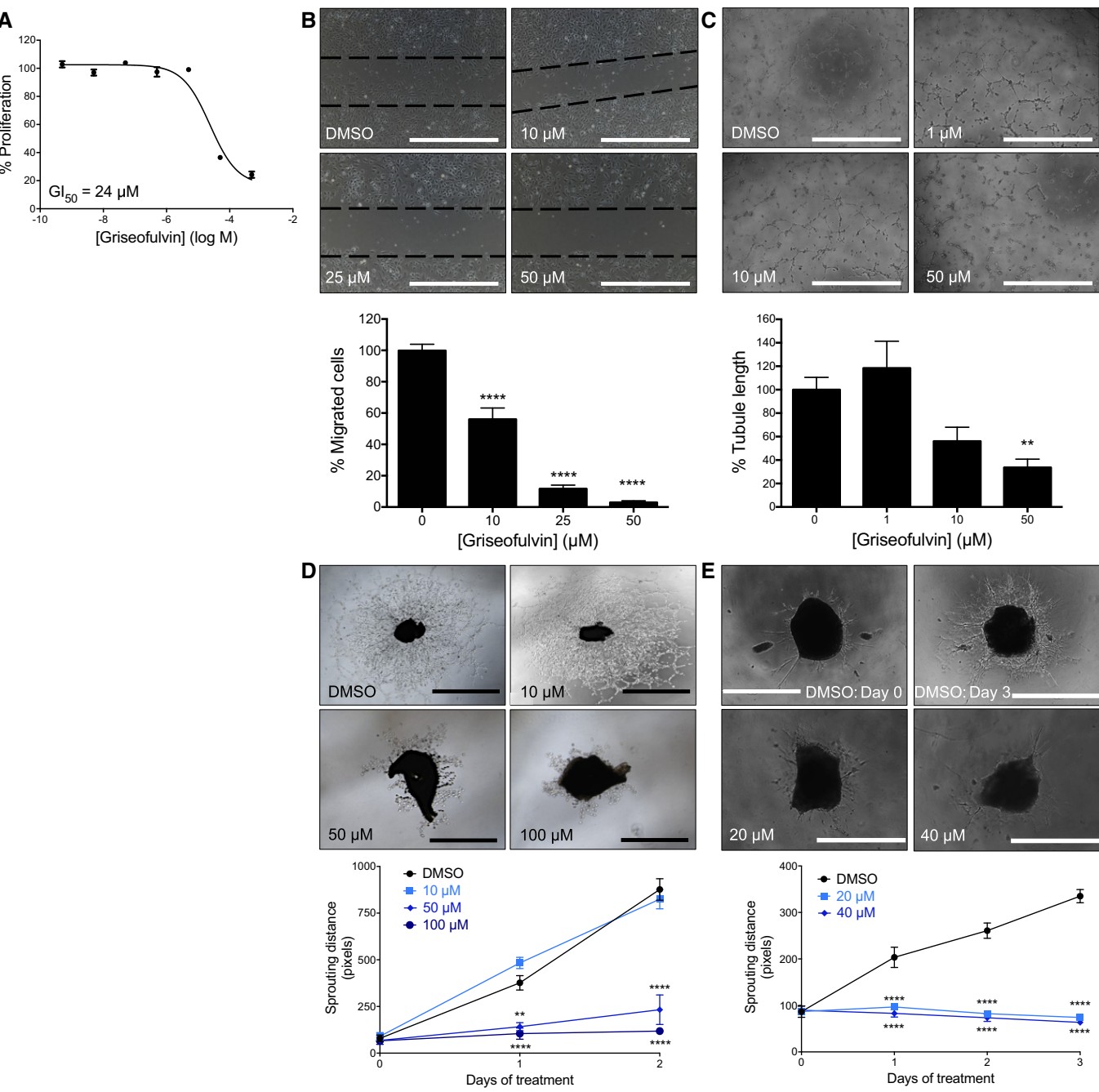

**Figure 4. Chemical inhibition of FECH inhibits angiogenesis *in vitro*.**

A The effect of griseofulvin, an FDA-approved drug that inhibits FECH activity, on proliferation of HRECs *in vitro*. (n = 3 per dose).

B Migration of HRECs in presence of griseofulvin was measured in a scratch-wound assay. ****$P$ = 0.0001, ANOVA with Dunnett's *post hoc* tests (n = 3 per group). Dotted lines indicate borders of scratch at time zero.

C Ability of HRECs treated with griseofulvin to form tubular structures in Matrigel was monitored and analyzed using ImageJ. **$P$ = 0.0082, ANOVA with Dunnett's *post hoc* tests (n = 6 per group).

D The mouse choroidal sprouting assay was used to further measure the antiangiogenic potential of griseofulvin. Representative phase contrast images and quantitative analysis of choroidal endothelial sprouting. **$P$ = 0.0018; ****$P$ = 0.0001 versus DMSO at same time point, two-way repeated measures ANOVA with Dunnett's *post hoc* tests (n = 3–4 per group).

E The *ex vivo* murine retinal angiogenesis assay was also used to measure the antiangiogenic potential of griseofulvin. Representative phase contrast images and quantitative analysis of retinal endothelial sprouting. ****$P$ = 0.0001 versus DMSO at same time point, two-way repeated measures ANOVA with Dunnett's *post hoc* tests (n = 5 per group).

Data information: Graphs show mean ± SEM, and the images are representative results from at least two independent experiments. Scale bars = 1 mm.

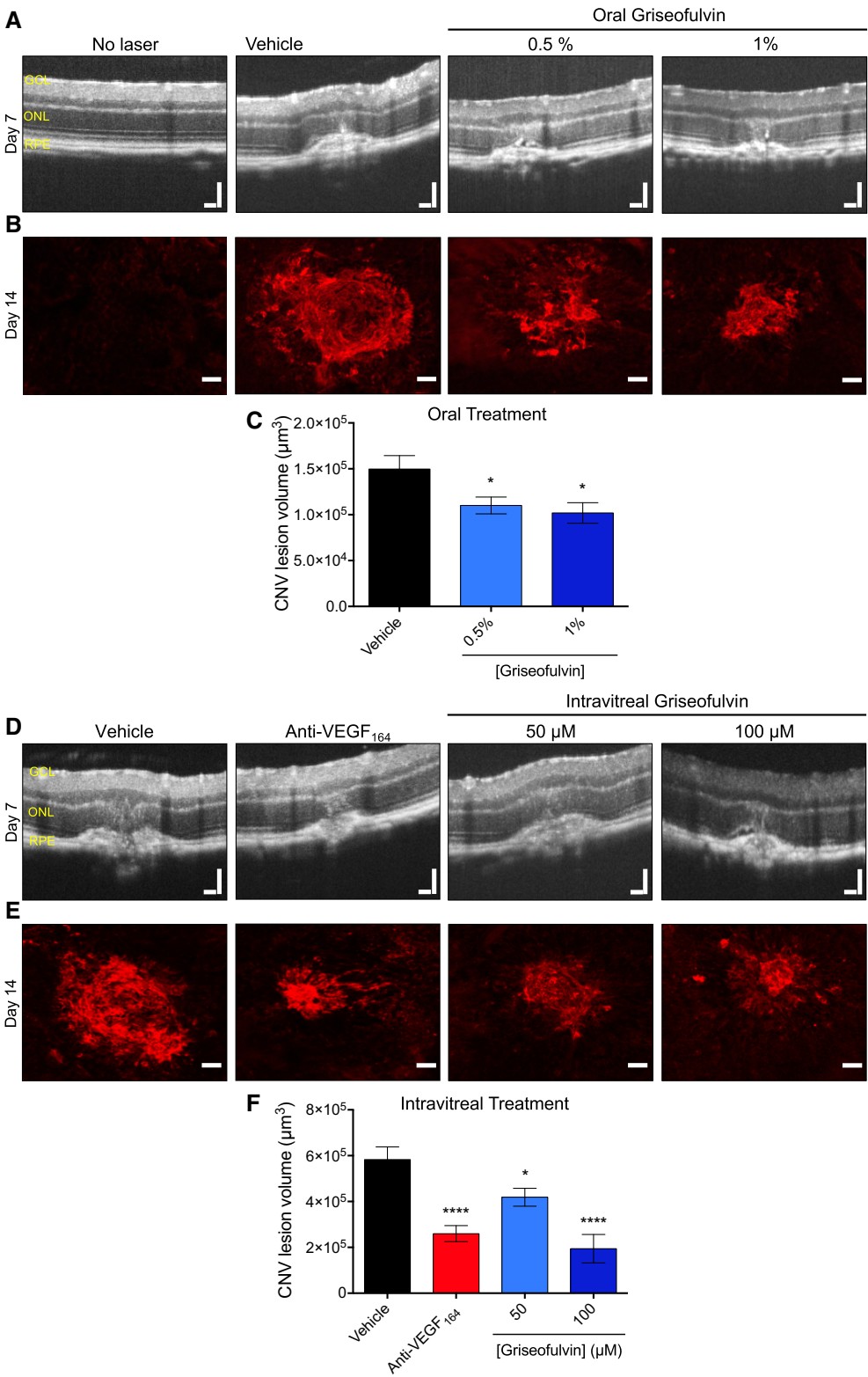

**Figure 5.**

requires a heme cofactor for its activity to produce the proangiogenic molecule nitric oxide (NO; Förstermann & Münzel, 2006). Immunoblot of HRECs treated with *FECH* siRNA showed significantly decreased levels of eNOS (Fig 6A). Further, NOS activity was decreased in *FECH* knockdown HRECs as compared to negative control siRNA-treated cells (Fig 6B). To assess hemylation of eNOS,

◀

**Figure 5.  Antifungal drug griseofulvin inhibits ocular neovascularization *in vivo*.**

A   Mice were fed *ad libitum* with 0.5 and 1% griseofulvin for 1 week prior to and throughout laser-induced choroidal neovascularization (L-CNV) development. CNV was monitored by optical coherence tomography (OCT) of mouse eyes *in vivo*. Retinal layers indicated: GCL, ganglion cell layer; ONL, outer nuclear layer; RPE, retinal pigment epithelium. CNV is observed as a broadening of the hyper-reflective tissues toward the bottom of the images, with corresponding deformation of the inner retinal layers.

B   Confocal imaging of CNV lesions *ex vivo* stained with agglutinin conjugated with Alexa Fluor 555 (red).

C   The lesion volumes were measured from confocal images. *$P$ = 0.048 and 0.013 versus vehicle, ANOVA with Dunnett's *post hoc* tests ($n$ = 15–17 eyes per group).

D   The effect of a single intravitreal injection of griseofulvin at time of laser treatment on choroidal neovascularization in the L-CNV model as monitored by OCT. Retinal layers indicated: GCL, ganglion cell layer; ONL, outer nuclear layer; RPE, retinal pigment epithelium.

E   Confocal imaging of CNV lesions *ex vivo* stained with agglutinin conjugated with Alexa Fluor 555 (red).

F   The CNV lesion volumes were measured from confocal images. *$P$ = 0.015; ****$P$ = 0.0001 versus vehicle, ANOVA with Dunnett's *post hoc* tests ($n$ = 11–13 eyes per group).

Data information: Graphs show mean ± SEM. Anti-VEGF$_{164}$ is a positive control antibody therapy. Scale bars for OCT images and immunostained choroids are 100 μm and 50 μm, respectively.

we used a hemin pull-down assay. Only hemoproteins in the *apo* form (enzyme lacking the heme cofactor) will bind to hemin agarose beads. Despite the overall decrease in eNOS levels, we observed that the remaining eNOS was in the *apo* form (Fig 6C) when HRECs were treated with NMPP, the specific inhibitor of FECH activity. As NO, the product of eNOS enzymatic activity, stabilizes hypoxia inducible factor (HIF) 1α (Sandau *et al*, 2001), we also monitored levels of this key mediator of angiogenesis after knocking down *FECH* in HRECS. *FECH* knockdown decreased protein levels HIF-1α (Fig 6D).

Since VEGF is a major proangiogenic stimulus, we monitored key events of VEGF signaling after knocking down *FECH*. We observed a profound decrease in (activating) phosphorylation of the major VEGF receptor VEGFR2 as well as total VEGFR2 protein levels in *FECH* knockdown cells when cultured in basal medium containing VEGF$_{165}$ (Fig 6E), although production of *VEGFA* mRNA was unchanged (Appendix Fig S2). Interestingly, levels of related receptors VEGFR1 and neuropilin 1 were not affected by FECH depletion (Fig 6E). The protein levels of eNOS and VEGFR2 were partially rescued after treating *FECH* knockdown cells with exogenously added hemin (a stable form of heme, the enzymatic product of FECH; Fig 6E). The decreased levels of eNOS, VEGFR2, and HIF-1α after *FECH* knockdown were not due to general translational inhibition: There was no difference in phosphorylation levels of eIF-2α in *FECH* siRNA-treated cells (Fig 6F).

## Discussion

Understanding the molecular events of pathological angiogenesis is key to developing novel therapeutics for neovascular eye diseases such as proliferative diabetic retinopathy, retinopathy of prematurity, and wet AMD. Currently, the drug pipeline for these diseases is dominated by anti-VEGF biologics (Kaiser, 2013). Even though these drugs have been successfully used to halt the progression of disease in wet AMD and proliferative diabetic retinopathy patients, there is a significant patient population (~30%) who are resistant to these treatments (Folk & Stone, 2010; Prasad *et al*, 2010). Hence diversification of the drug pipeline with novel therapeutic agents with different mechanisms of action is required (Smith & Kaiser, 2014). Toward this end, we used a forward chemical genetics approach using an antiangiogenic natural product, cremastranone, to uncover new drug targets for angiogenesis.

Using a series of biochemical, *in vitro*, and *in vivo* studies, we have determined that FECH is a key protein involved in ocular

angiogenesis that can be exploited for developing novel antiangiogenic drugs. A photoaffinity-based chromatographic technique identified FECH as a protein binding partner of cremastranone. Although, like other natural products, cremastranone likely exhibits polypharmacology (Sulaiman *et al*, 2014), our data indicate that it exerts its antiangiogenic activity at least partially through inhibition of FECH activity.

Interestingly, FECH is a *bona fide* mediator of angiogenesis. Lack of FECH activity caused inhibition of angiogenesis both *in vitro* and *in vivo*. More importantly, only retinal and choroidal microvascular endothelial cell proliferation was inhibited *in vitro* while other ocular cell types tested did not show significant decreases in cell proliferation after *FECH* knockdown. Even macrovascular HUVECs were not as profoundly affected by FECH depletion as the microvascular HRECs and Rf/6a cells. These data reveal that microvascular endothelial cells are particularly sensitive to FECH depletion. The lack of cytotoxic effects of FECH inhibition leads to consideration of FECH as a novel therapeutic target for ocular neovascular disease, possibly with minimal side effects. Supporting this latter assertion, in the genetic disease erythropoietic protoporphyria (EPP), FECH activity is markedly reduced, but EPP patients infrequently present severe symptoms apart from skin photosensitivity (Lecha *et al*, 2009). It may be possible to tune FECH activity to minimize side effects while maximizing antiangiogenic effects: Heterozygous *Fech*^m1Pas mice do not display an EPP phenotype, despite reduced FECH function (Boulechfar *et al*, 1993), but do show reduced choroidal neovascular response (this study). Neovascularization in human EPP patients has not been studied extensively, although a single case report describes an ocular phenotype of idiopathic optic neuropathy in an EPP patient, not conclusively related to the disease (Tsuboi *et al*, 2007).

Excitingly, we have shown that the FDA-approved antifungal drug and FECH inhibitor griseofulvin inhibited ocular angiogenesis in the L-CNV mouse model when administered orally. Griseofulvin has been used widely to treat fungal infections and is taken orally, often for months or years (Petersen *et al*, 2014; Liu *et al*, 2015). An off-target side effect of this therapy is that griseofulvin causes the formation of NMPP, along with other alkylated porphyrins, primarily in the liver (Liu *et al*, 2015). NMPP in turn acts as an active-site inhibitor of FECH (Cole & Marks, 1984). As with genetic reduction in function of FECH, apart from skin photosensitivity, no other major, common side effects are reported with systemic griseofulvin treatment of humans (Elewski & Tavakkol, 2005; Grover *et al*, 2012).

The fact that griseofulvin-fed mice showed decreased ocular neovascularization as compared to control mice is important as

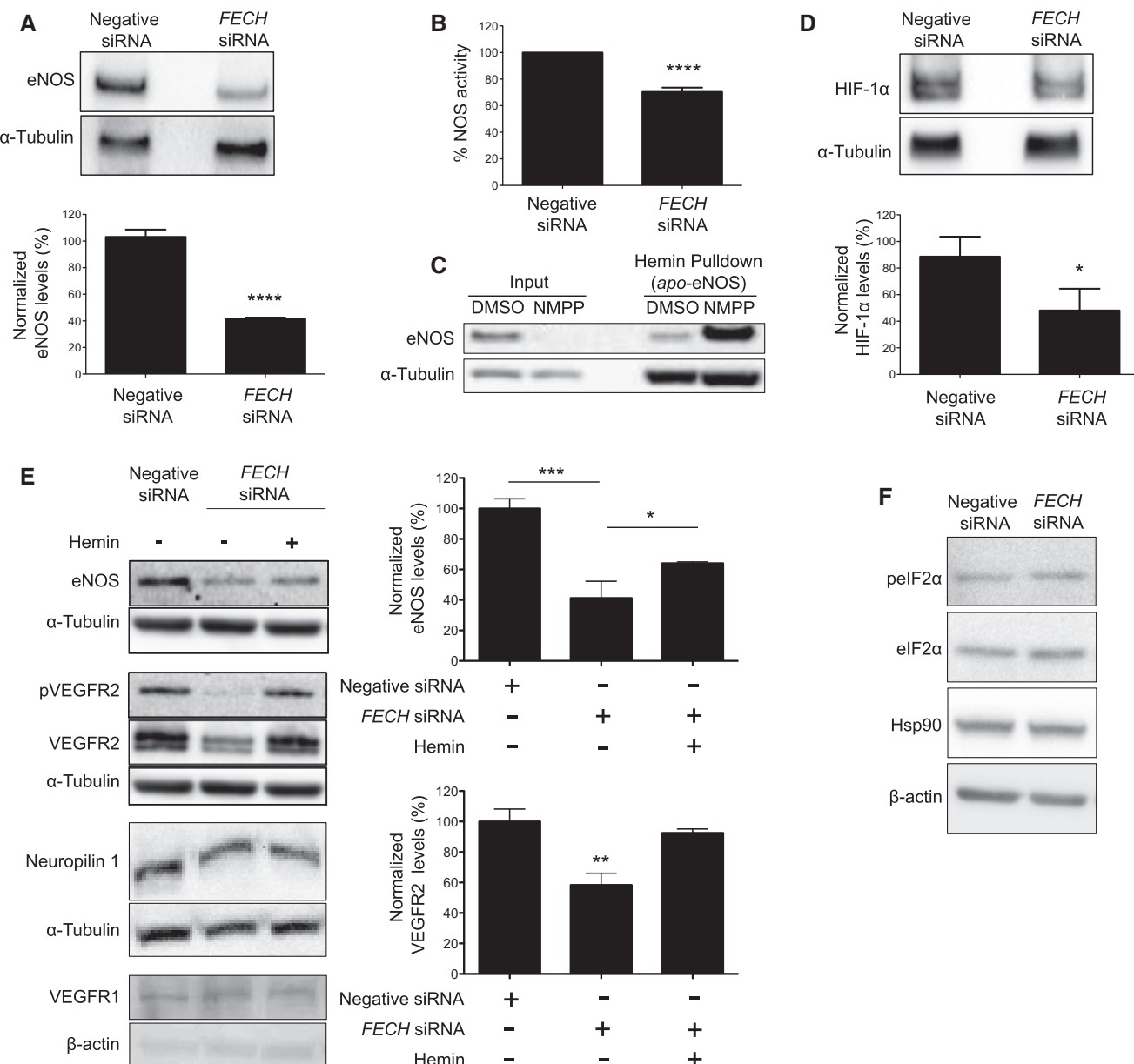

**Figure 6.  FECH depletion suppresses VEGFR2 via eNOS and HIF-1α.**

A   Protein levels of eNOS were measured by immunoblot after knocking down *FECH* in HRECs. ****$P$ = 0.0001, two-tailed unpaired Student's $t$-test ($n$ = 3 per group).

B   Production of NO was monitored using a diaminofluorescein-2-diacetate-based fluorometric assay. ****$P$ = $8 \times 10^{-10}$, two-tailed unpaired Student's $t$-test ($n$ = 6 per group).

C   Immunoblots from hemin pull-down assay (to detect *apo*-eNOS) using proteins extracted from HRECs treated with DMSO or 10 μM NMPP.

D   HIF-1α was measured by immunoblot after knocking down *FECH* in HRECs. *$P$ = 0.043, two-tailed unpaired Student's $t$-test ($n$ = 3 per group).

E   Protein levels of eNOS, neuropilin, VEGFR1, and VEGFR2, plus phosphorylation of VEGFR2 at Tyr1175 in HRECs after treatment with negative control or *FECH* siRNAs and/or hemin (0.1 μM) in complete medium were monitored by immunoblot. ***$P$ = 0.0005; *$P$ = 0.0426 (for eNOS); **$P$ = 0.0069 versus negative siRNA (for VEGFR2), ANOVA with Tukey's *post hoc* tests ($n$ = 3 per group).

F   Phosphorylation of eIF2α at Ser51 and protein levels of Hsp90 were measured by immunoblot after knocking down *FECH* in HRECs.

Data information: The figures are representative from at least three independent experiments and the graphs show mean ± SEM.
Source data are available online for this figure.

currently there are no oral drugs available for the treatment of ocular neovascularization. The current approved drugs (anti-VEGF biologics) are all delivered intravitreally. If side effects were manageable, an oral therapy would be appealing. However, since griseofulvin was also effective when delivered intravitreally, combination intravitreal therapies of this drug with anti-VEGF biologics are also an interesting possibility to increase efficacy and decrease side effects of existing treatments. Moreover, griseofulvin offers the

promise of low-cost treatment if shown efficacious in clinical trials. There are also prospects for novel FECH inhibitors as therapy: We have developed cremastranone analogs that may also act through FECH (Basavarajappa *et al*, 2015), and recently, the drugs vemurafenib (Savitski *et al*, 2014) and salicylic acid (Gupta *et al*, 2013), as well as a lipid-derived probe compound (Niphakis *et al*, 2015), were all shown to bind to FECH, suggesting that this protein has many potential binding clefts and is therefore likely druggable with novel compounds.

FECH is the only known enzyme in humans to synthesize heme, and any loss of activity of this enzyme might result in intracellular heme deficiency (Dailey *et al*, 2000). Heme serves as either a prosthetic group for many enzymes or as a signaling molecule regulating expression of genes involved in cell differentiation, proliferation, and signal transduction (Mense & Zhang, 2006). *FECH* knockdown decreased protein levels of eNOS, HIF-1α and VEGFR2, key players promoting angiogenesis, in HRECs. As eNOS is a heme-dependent enzyme, *FECH* knockdown directly affects the stability and activity of eNOS. The product of the eNOS enzyme, NO, induces angiogenesis (Ziche & Morbidelli, 2000), partly by stabilizing HIF-1α under normoxia (Sandau *et al*, 2001). This explains the logic behind the decreased protein levels of HIF-1α in *FECH* knockdown cells with decreased eNOS activity. HIF-1α is a transcription factor that regulates the expression of a range of proangiogenic proteins. VEGFR2, the major tyrosine kinase receptor of VEGF$_{165}$, is upregulated under hypoxia (Tuder *et al*, 1995; Takagi *et al*, 1996). Thus, the decreased levels of VEGFR2 in HRECs after *FECH* siRNA treatment can be linked to decreased HIF-1α. Taken together, low protein levels of VEGFR2 (and thus angiogenesis) in FECH-deficient HRECs can be directly attributed to loss of heme in cells and the resultant decrease in levels of eNOS enzyme. VEGFR2 depletion in turn further suppresses eNOS levels. The rescue of VEGFR2 and eNOS expression by addition of exogenous hemin, a stable form of heme, further supports this mechanism.

Our findings reveal a previously undocumented, central role of FECH in ocular angiogenesis. They provide a rationale for clinical testing of griseofulvin in neovascular eye disease. In addition, it will be valuable to develop novel, FECH-targeted therapies for treating the debilitating ocular diseases caused by neovascularization.

## Materials and Methods

### Materials

EBM-2 and IMDM growth media were purchased from Lonza (Walkersville, MD, USA). HRECs, BMECs, and Attachment Factor were purchased from Cell Systems (Kirkland, WA, USA). Clonetics® HUVECs were purchased from Lonza. All endothelial cells were used between passages 5 and 8. Endothelial growth medium (EGM-2) was prepared by mixing the contents of an EGM-2 "Bullet Kit" (Cat no. CC-4176) with endothelial basal medium (EBM; Lonza). Rf/6a cells were obtained from ATCC (Manassas, VA, USA) and grown in EMEM. Y-79 cells were grown in RB medium (Basavarajappa *et al*, 2015), while 92-1 (Basavarajappa *et al*, 2015) and ARPE-19 (ATCC) cells were grown in RPMI and DMEM, respectively, supplemented with 10% FBS and 1% penicillin–streptomycin as described earlier (Basavarajappa *et al*, 2015); identity was

confirmed by STR profiling and cells were monitored regularly for mycoplasma contamination. Click-iT TUNEL Alexa Fluor 594 imaging assay kit (Cat no. C10246) was purchased from Molecular Probes (Eugene, OR, USA). Monoclonal antibodies against α-tubulin (DM1A) and β-actin (AC40), protoporphyrin IX, 5-aminolevulinic acid, hemin, hemin agarose beads, griseofulvin, and L-arginine were purchased from Sigma-Aldrich (St. Louis, MO, USA). *N*-methyl protoporphyrin (NMPP) and the primary antibodies against FECH (A-3 and C20) were obtained from Santa Cruz (Santa Cruz, CA, USA). Antibodies against cleaved caspase 3 (5A1E), phospho-VEGFR2 (Tyr 1175) (19A10), phospho-eIF2α (Ser 51) (9721), eIF2α (9722), and eNOS (49G3) were from Cell Signaling (Danvers, MA, USA). Anti-HIF-1α (241809), VEGFR1 (AF321), and anti-VEGF$_{164}$ (AF-493-NA) antibodies were purchased from R&D systems (Minneapolis, MN, USA). The antibody against neuropilin 1 (ab81321) was purchased from Abcam (Cambridge, MA, USA). Secondary antibodies were from Thermo Fisher Scientific (Pittsburgh, PA, USA). The Trizol, TaqMan probes, qPCR master mix, and 5-ethynyl-2′-deoxyuridine (EdU) incorporation assay kit were procured from Life Technologies (Carlsbad, CA, USA). The iScript reverse transcriptase was from Bio-Rad (Hercules, CA, USA). AbD Serotec (Kidlington, UK) was the source of the alamarBlue, while BD Biosciences (San Jose, CA, USA) supplied the Matrigel. 4,5-Diaminofluorescein diacetate (DAF-2 diacetate) was purchased from Cayman Chemicals (Ann Arbor, MI, USA). ECL Prime Western blotting detection reagent was purchased from GE Healthcare (Buckinghamshire, UK).

### Preparation of photoaffinity reagents

Synthesis and characterization of affinity reagents **2** and **3** was performed as described (Lee *et al*, 2016). Compound **2** contains a trimethoxy derivative of cremastranone that is more synthetically tractable, but which maintains good cellular potency and selectivity, both as homoisoflavonoid (Basavarajappa *et al*, 2015) and when incorporated into the affinity reagent (Lee *et al*, 2016). Compounds **1** and **4** were synthesized as described (Basavarajappa *et al*, 2014; Lee *et al*, 2014), with purity > 95%. For pull-downs, Neutravidin agarose beads (1 ml of 50% slurry) were washed three times in buffer A containing 25 mM Tris–HCl pH 7.4, 150 mM NaCl, 2.5 mM sodium pyrophosphate, 1 mM phenylmethylsulfonyl fluoride (PMSF), 0.1 mM sodium orthovanadate, 10 μg/ml aprotinin, and 10 μg/ml leupeptin. The beads were then incubated with 100 μM affinity or control reagents **2** or **3** overnight at 4°C with rotation. The beads were blocked using 1 mM biotin solution prepared in buffer A for 1 h followed by incubation with 1 mg/ml cytochrome *c* solution for 1 h at 4°C. The beads were then washed three times with buffer A and resuspended in 1 ml.

### Photoaffinity pull-down experiments

Flash-frozen porcine brain (20 g) obtained from the Purdue-Indiana University School of Medicine Comparative Medicine Program was homogenized in 50 ml buffer A using a tissue homogenizer. Porcine brain was chosen as it is readily available and a rich source of protein. The homogenate was centrifuged at 2,000 *g* for 5 min. The supernatant (S1) was then homogenized by 50 strokes of a Dounce homogenizer followed by 10 min sonication with amplitude of 60% in cycles of 10 s sonication on and 40 s sonication off (Q125 from

QSonica, Newtown, CT, USA). The lysate was then centrifuged at 11,000 $g$ for 30 min. The resulting supernatant (S2) and pellet (P2) fractions were both collected. The P2 pellet was resuspended in buffer B: 1% Triton X-100 + buffer A and then homogenized by 25 strokes of a Dounce homogenizer and centrifuged at 11,000 $g$ for 30 min; supernatant (S3) was collected. Both S2 and S3 supernatants were equally divided, and each fraction was incubated with 500 μl photoaffinity or control reagent conjugated to Neutravidin beads for 75 min at 4°C with shaking.

The beads were collected by centrifugation at 500 $g$ for 5 min, then resuspended in 1 ml of buffer B and irradiated with 365 nm UV light (Mercury bulb H44GS100 from Sylvania in a Blak-Ray 100A long-wave UV lamp with output of 25 mW/cm$^2$ at sample distance) in a 60-mm Petri dish for 30 min at 4°C. The beads were then washed two times in buffer B, followed by three washes in high-salt buffer containing 25 mM Tris–HCl pH 7.4, 350 mM NaCl, 1% Triton X-100, and 1 mM PMSF. The beads were then washed again in salt-free buffer containing 25 mM Tris–HCl, 1% Triton X-100, and 1 mM PMSF. After 5-min incubation, the beads were collected and any residual liquid was removed using a Hamilton syringe. The Neutravidin beads were then heated in 300 μl of 2× SDS–PAGE gel loading dye containing 30 μl of 2-mercaptoethanol for 10 min at 70°C to release the bound proteins. After heating, the contents were allowed to cool and after a quick spin the eluate was collected using a Hamilton syringe. The eluates were then analyzed in 4–20% gradient SDS–PAGE, and the protein bands were visualized using silver staining (Corson *et al*, 2011). The protein bands pulled down specifically by photoaffinity reagent were excised from the silver-stained SDS–PAGE gel and analyzed by mass spectrometry (IUSM Proteomics Core). Using Sequest™ algorithms and the swine database (UniProt), the identities of the pulled-down proteins were confirmed (Appendix Fig S1).

For competition experiments, S2 and S3 supernatants were incubated with affinity reagent-Neutravidin beads in the presence of 1 mM cremastranone isomer SH-11052 (**4**) (Basavarajappa *et al*, 2014) and then processed as described above.

## Recombinant FECH

Recombinant human FECH protein was purified as described previously (Dailey *et al*, 1994). Briefly, *Escherichia coli* JM109 cells transformed with plasmid pHDTF20 encoding recombinant human *FECH* were grown in Circlegrow medium containing 100 μg/ml ampicillin for 20 h at 30°C. The cells were harvested and resuspended in solubilization buffer (50 mM Tris-MOPS pH 8.0, 1% sodium deoxycholate, 100 mM KCl, and 1 mM PMSF). The cell suspension was sonicated and then ultracentrifuged at 45,000 $g$ for 30 min. The supernatant was subjected to cobalt-affinity chromatography, and the column was washed with solubilization buffer containing 20 mM imidazole. The protein was eluted with 250 mM imidazole in solubilization buffer. The protein eluate was then dialyzed in solubilization buffer containing 10% glycerol before storage at 4°C. Recombinant protein (200 μg) was used in pull-down experiments as above.

## Immunoblot assay

Immunoblots were performed as described previously (Basavarajappa *et al*, 2014). Briefly, cell lysates were prepared by incubating the cells for 10 min at 4°C in NP-40 lysis buffer (25 mM HEPES pH 7.6, 150 mM NaCl, 1% NP-40, 10% glycerol, 1 mM sodium orthovanadate, 10 mM NaF, 1 mM PMSF, 10 μg/ml aprotinin, 1 μM pepstatin, 1 μM leupeptin) and then centrifuged at 12,000 $g$ for 15 min at 4°C. Supernatant was collected, and protein concentration was determined using a Bradford assay. Equal amounts of total protein (40 μg) from each sample were resolved by 10% SDS–PAGE and transferred onto PVDF membranes. Proteins were immunoblotted with antibodies against FECH (clone A-3; 1:1,000 dilution), α-tubulin (1:1,000), VEGFR1 (1:500), phospho-VEGFR2 (1:500), VEGFR2 (1:500), phospho-eIF2α (1:500), eIF2α (1:500), HIF-1α (1:500), neuropilin 1 (1:500), β-actin (1:1,000), or eNOS (1:500). All of the dilutions were made in Tris-buffered saline-0.05% Tween-20 buffer containing 2% bovine serum albumin (BSA).

## siRNA knockdown of *FECH* in cells

Cells were grown in 6-well plates until 80% confluency was achieved. Lipofectamine RNAiMAX reagent (7.5 μl, Life Technologies) mixed with 30 pmol of siRNAs was added to each well according to the protocol recommended by the manufacturer. For *FECH* knockdown, 15 pmol each of two siRNAs (SASI_Hs01_00052189 and SASI_Hs01_00052190; Sigma) was used, and for negative control, MISSION® siRNA Universal negative control was used. Fresh EGM-2 medium was added to the plate 24 h after transfection, and cells were used 48 h after transfection for further experiments except for the proliferation time course, for which 24 h after transfection the cells were trypsinized and seeded in a 96-well plate.

## Cell proliferation assay

Proliferation of cells was monitored as described before (Basavarajappa *et al*, 2014). Briefly, 2,500 cells (after *FECH* knockdown in some experiments) in 100 μl of growth medium were plated in each well of 96-well clear-bottom black plates and incubated for 24 h. Griseofulvin, NMPP, or DMSO vehicle (final DMSO concentration of 1%) was added, and the plates were incubated for 44 h in 100 μl media at 37°C and 5% CO$_2$. For washout experiments, 1,500 cells were initially plated, and drug-containing medium was removed after 48 h and cells incubated in fresh growth medium for a further 24 or 48 h. AlamarBlue reagent (11.1 μl) was added to each well of the plate and 4 h later fluorescence readings were taken at excitation and emission wavelengths of 560 nm and 590 nm, respectively, using a Synergy H1 plate reader (BioTek, Winooski, VT, USA). For dose–response experiments, GI$_{50}$ was calculated using GraphPad Prism v. 7.0.

## Migration assay

The migration of HRECs was monitored as described before (Basavarajappa *et al*, 2014). Briefly, HRECs were grown until confluency in 6-well plates and then serum starved overnight in EBM-2 medium. Using a sterile 10-μl tip, a scratch was introduced in each well and fresh EGM-2 medium containing DMSO or different concentrations of compounds was added to the wells. For knockdown experiments, the scratch was introduced 48 h after transfection and fresh medium was added to the wells. Photographs of the

wells were taken at different time points, and the number of migrated cells into the scratched area was manually counted.

## In vitro Matrigel tube formation assay

The ability of HRECs and Rf/6a cells to form tubes *in vitro* was monitored as described before (Basavarajappa *et al*, 2015). Briefly, cells were treated with the indicated concentrations of compounds or DMSO or siRNAs for 48 h and then 15,000 cells in 100 µl of growth medium containing siRNAs, DMSO, or compounds were added to each well of a 96-well plate that was pre-coated with 50 µl of Matrigel basement membrane. Photographs of each well at different time points were taken to measure the tube formation using the Angiogenesis Analyzer plugin in ImageJ software (v.1.48; http://image.bio.methods.free.fr/ImageJ/?Angiogenesis-Analyzer-for-ImageJ.html).

## Animals

All animal experiments were approved by the Indiana University School of Medicine Institutional Animal Care and Use Committee and followed the guidelines of the Association for Research in Vision and Ophthalmology Statement for the Use of Animals in Ophthalmic and Visual Research. Wild-type female C57BL/6J mice, 6–8 weeks of age or timed pregnancies, were purchased from Jackson Laboratory (Bar Harbor, ME) and housed under standard conditions (Wenzel *et al*, 2015). $Fech^{m1Pas}$ mice (Tutois *et al*, 1991) were purchased from Jackson Laboratory on a BALB/c background and backcrossed into C57BL/6J; mixed-sex littermates from a subsequent brother–sister mating were used for experiments at 6–8 weeks of age. Mice were anesthetized for all procedures by intraperitoneal injections of 80 mg/kg ketamine hydrochloride and 10 mg/kg xylazine. Sample sizes were based on power analyses, and treatments were randomly assigned by cage.

## L-CNV model

L-CNV was generated as described previously (Lambert *et al*, 2013; Poor *et al*, 2014; Sulaiman *et al*, 2015). Studies were powered to have an 80% chance of detecting effect size differences of 50%, assuming 30% variability, α = 0.05. Briefly, eyes were dilated using 1% tropicamide, then underwent laser treatment using 50 µm spot size, 50 ms duration, and 250 mW pulses of an ophthalmic argon green laser, wavelength 532 nm, coupled to a slit lamp. Where indicated, intravitreal injections of PBS vehicle, siRNA (1.25 µM final intravitreal concentration), griseofulvin (50, 100 µM, final intravitreal concentrations), anti-VEGF$_{164}$ antibody (0.2, 1, 5 ng delivered), or combinations of griseofulvin and anti-VEGF$_{164}$ antibody, as indicated, were given in a 0.5 µl volume at the time of laser treatment. Eyes were numbed with tetracaine solution before the injection, and triple antibiotic ointment was used immediately after the injection to prevent infection. A masked researcher undertook imaging and analysis to avoid bias. One week after laser treatment, mice underwent optical coherence tomography using a Micron III imager (Phoenix Research Labs, Pleasanton, CA, USA) and CNV lesions were quantified as ellipsoids as described (Sulaiman *et al*, 2015). Two weeks after laser treatment, eyes were enucleated and fixed, choroidal flat mounts prepared, and vasculature stained with rhodamine-labeled *Ricinus communis* agglutinin I (Vector Labs,

Burlingame, CA, USA), followed by confocal *Z*-stack imaging (LSM 700, Zeiss, Thornwood, NY, USA) to estimate lesion volume. The sum of the stained area in each section, multiplied by the distance between sections (3 µm), gave the CNV lesion volume.

## Immunostaining

Human donor eyes from wet AMD patients or age-matched controls were obtained from the National Disease Research Interchange (NDRI; Philadelphia, PA) with full ethical approval for use in research. NDRI obtains informed consent for all donor material (http://ndriresource.org/for-donors-patients) and all procedures conform to the principles set out in the WMA Declaration of Helsinki and the Department of Health and Human Services Belmont Report. All sample tissues were anonymized prior to receipt in the laboratory. Mouse eyes were harvested 14 days after L-CNV induction. The eyes were enucleated and fixed in 4% paraformaldehyde/PBS overnight. The anterior segment, lens, and vitreous humor were removed, and the posterior eye cups were prepared for standard paraffin sections or retinal flat mounts. Deparaffinized sections were treated with rodent deblocker (Biocare Medical, Concord, CA, USA) for antigen retrieval. The sections or flat mounts were washed with PBS then permeabilized with 0.3% Triton X-100 and nonspecific binding blocked by 10% normal goat serum plus 1% BSA in PBS. They then received primary antibody (polyclonal anti-FECH (clone C20)) at 1:500 for 16 h at 4°C. After primary incubation, sections or whole mounts were washed and incubated for 1.5 h at room temperature with secondary antibody (Cy3-conjugated goat anti-rabbit IgG, 1:600) at 4°C with 0.1% Triton X-100. We used a vascular-specific lectin (*Ricinus communis* agglutinin I; Vector Laboratories, Inc.) conjugated to Alexa Fluor 488 to label the retinal vasculature. This was incubated for 30 min at room temperature in 1:400 of 10 mM HEPES plus 150 mM NaCl and 0.1% Tween-20. After washing, specimens were mounted in aqueous mounting medium (VectaShield; Vector Laboratories, Inc.) and coverslipped for observation by confocal microscopy and quantification where noted by a masked investigator using ImageJ software. All microscopic images in a given panel were acquired with identical exposure settings.

## Choroidal sprouting assay

Choroidal sprouting was assessed as described (Sulaiman *et al*, 2016). Briefly, pieces of choroid–sclera dissected from mouse eyes were embedded in Matrigel and grown in EGM-2 medium plus antibiotics for 72 h to allow sprouting to initiate. The indicated concentrations of griseofulvin (in DMSO, final DMSO concentration 1%) were added and growth allowed to proceed for 48 h. Images were taken and growth quantified by measuring the distance from the edge of the choroidal piece to the growth front in four directions per sample.

## Retinal endothelial cell sprouting assay

An *ex vivo* murine retinal angiogenesis assay (EMRA) was performed according to the previously published protocol (Rezzola *et al*, 2013) with slight modifications. Briefly, retinal fragments were isolated from adult C57BL/6J mice (3.5–4.5 weeks old) and embedded in fibrin gel [containing bovine fibrinogen (4 mg/ml), aprotinin (5 µg/µl), and bovine thrombin (500 mU/ml)] in serum-free DMEM

and allowed to gel at 37°C. After clotting for 20–30 min, 400 μl of DMEM containing 10% FBS and human VEGF$_{165}$ (100 ng/well) was added and replaced after 3 days. After day 7, medium was replaced with DMEM containing 2% FBS plus VEGF$_{165}$ (100 ng/well) and treated with griseofulvin (20 and 40 μM) or vehicle control (DMSO, 0.5% final concentration). Endothelial sprouting was monitored and photographed daily until day 3 after treatment. Image analysis of the retinal endothelial sprouting was performed manually using ImageJ software, and quantification was performed of least six regions of interest per treatment per time point. Endothelial cell sprouting was represented as sprouting distance in pixels.

### Griseofulvin feeding

Mice were fed griseofulvin for a total of 3 weeks, with chow changed every 2–3 days. Standard mouse chow (5 g/mouse/day) was mixed in water (2.2 ml H$_2$O/gram chow), soaked for 15 min, and then mashed. Griseofulvin doses were prepared at 0.0% (control), 0.5%, and 1.0% with 0.0, 0.5, and 1.0 g griseofulvin : 10 ml corn oil : 100 g mouse chow ratio, respectively. Both 0.5 and 1.0% doses were expected to substantially inhibit FECH and induce a protoporphyria-like phenotype, based on previously published work (Lochhead *et al*, 1967; Holley *et al*, 1990; Martinez *et al*, 2009). During treatment, the mice were examined and weighed 3 times/week (Fig EV5A). On day 8, mice underwent L-CNV as above and were imaged by OCT at day 15 and day 22, at which time they were euthanized. The eyes were enucleated and flat mounts prepared as described above. The livers were dissected out and weighed (Fig EV5B).

### NOS assay

NOS activity was measured in cells as described before (Räthel *et al*, 2003). Briefly, HRECs (10,000 cells/well) were seeded in 96-well clear-bottom black plates and incubated for 24 h at 37°C and 5% CO$_2$. Cells were washed with PBS and incubated for 5 min at 37°C with 100 μl of 100 μM L-arginine prepared in PBS. Subsequently, 100 nM of DAF-2 diacetate was added to each well and fluorescence readings were taken at excitation and emission wavelengths of 495 and 515 nm, respectively, using the Synergy plate reader (Biotek).

### Hemin pull-down

HRECs were grown in EGM-2 medium in 10-cm plates until they reached ~50% confluency. Cells were treated with 10 μM NMPP or DMSO control for 1 week. Medium was changed every 2 days with treatments added to the fresh medium every time. Cells were lysed with NP-40 lysis buffer (20 mM Tris, pH 8.0, 150 mM NaCl, 1% NP-40, 20 μM leupeptin, 1 mM PMSF, 1 mM NaF, 1 mM β-glycerophosphate, 2 mM sodium orthovanadate, 2 mM EDTA, 10% glycerol) and then centrifuged at 14,000 *g* for 10 min at 4°C. Supernatant was collected and samples were pre-cleared by incubation with Neutravidin beads for 1 h at 4°C followed by centrifugation at 500 *g* for 5 min at 4°C. Supernatant was collected, and protein concentration was determined using a Bradford assay. Equal amounts of total protein (40 μg) from each sample were incubated with ~50 μl hemin agarose beads, pre-washed three times with lysis buffer, for 1 h at 4°C.

The beads were then washed two times in NP-40 buffer, followed by two washes in high-salt buffer containing 20 mM Tris–HCl pH 8.0, 350 mM NaCl, and 1 mM PMSF. The beads were then washed once with a very high-salt buffer containing 20 mM Tris–HCl pH 8.0, 500 mM NaCl, and 1 mM PMSF. The beads were then washed again in salt-free buffer containing 20 mM Tris–HCl and 1 mM PMSF. After 5-min incubation, the beads were collected and any residual liquid was removed using a Hamilton syringe. The hemin agarose beads were heated in 30 μl of 2× SDS–PAGE gel loading dye containing 30 μl of 2-mercaptoethanol for 10 min at 70°C to release the bound proteins. After heating, the contents were allowed to cool, and after a quick spin, the eluate was collected using a Hamilton syringe. Eluates were separated in a 4–20% gradient SDS–PAGE and then transferred onto PVDF membranes. Proteins were immunoblotted with antibodies against α-tubulin (1:1,000) or eNOS (1:500). All dilutions were made in Tris-buffered saline–0.05% Tween-20 buffer containing 2% bovine serum albumin (BSA).

### PPIX buildup assay

HRECs were grown in a 6-well plate until confluent. Cells were serum starved overnight in EBM-2 medium. Fresh EGM-2 medium containing DMSO or cremastranone (**1**) was added to cells, and they were incubated at 37°C for 1 h followed by the addition of 1 mM 5-ALA to increase flux through the heme biosynthetic pathway. After 3 h of incubation in the dark at 37°C, the cells were trypsinized and lysed in buffer containing 25 mM HEPES–NaOH pH 7.4, 150 mM NaCl, 1% NP-40, 10% glycerol, and 1 mM PMSF. The cell lysates were incubated in the dark at 4°C for 20 min on a shaker and centrifuged at 12,000 *g* for 15 min. Supernatants were collected for analysis. In a 384-well black plate, 20 μl of supernatant was mixed with 20 μl of 1:1 solution of 2 M perchloric acid and methanol. After 5 min of incubation, fluorescence readings were taken at excitation and emission wavelengths of 407 and 610 nm using the Synergy plate reader.

### Iron chelation

Compounds (cremastranone, EDTA, or deferoxamine) or DMSO (1 μl) were incubated with 2.5 mM freshly prepared ferrous ammonium sulfate in a final volume of 100 μl for 5 min at 37°C. Ferrozine (100 μl of 2.5 mM solution) was added to the wells, and spectrophotometric readings were taken at 562 nm using the Synergy plate reader. Decrease in absorbance readings at 562 nm represented the degree of iron chelation.

### Apoptosis assays

The caspase-3 immunofluorescence assay was performed as described previously (Basavarajappa *et al*, 2014). Briefly, cells were plated on coated coverslips and incubated in EGM-2 medium overnight before treating with siRNAs. After 24 h of transfection, the cells were fixed in 4% paraformaldehyde and permeabilized using 0.5% Triton X-100 solution prepared in PBS. The cells were incubated with cleaved caspase-3 antibody (1:200 dilution) overnight at 4°C. Dylight 488-conjugated goat anti-rabbit secondary antibody (1:400) was used to probe the cleaved caspase-3 antibody. The

coverslips were mounted using Vectashield mounting medium containing DAPI for nuclear staining. The cells were imaged using an LSM 700 confocal microscope.

The TUNEL assay was carried out as described previously (Basavarajappa *et al*, 2015). Briefly, cells (25,000 per coverslip) were seeded on coverslips and 24 h later the cells were transfected with siRNAs or treated with the indicated compound concentrations for 24 h. Cells were then fixed in 4% paraformaldehyde for 20 min and permeabilized using 0.25% Triton X-100 prepared in PBS. Then, apoptotic cells were visualized using the Click-iT TUNEL assay kit as per the manufacturer's instructions, with DAPI counter-stain. The percentage of apoptotic cells was counted on three low-power fields per coverslip using ImageJ software.

## qRT–PCR

The assay was performed as described previously (Basavarajappa *et al*, 2014). RNA was extracted from cells treated as indicated using Trizol. cDNA was synthesized from 500 ng RNA using random primers and iScript reverse transcriptase. qPCR was performed in 10 µl volumes in a 384-well plate, with Fast Advanced Master Mix and TaqMan probes on a ViiA7 thermal cycler (Life Technologies). Primer/probesets used were as follows: *FECH* (Hs01555261_m1), *HIF1A* (Hs00153153_m1), *NOS3* (Hs01574659_m1), *VEGFA* (Hs00900055_m1), *VEGFR2* (Hs00911700_m1), and housekeeping controls *GAPDH* (Hs99999905_m1), *HPRT* (Hs02800695_m1), and *TBP* (Hs00427620_m1). The data were analyzed using the $\Delta\Delta C_t$ method (Livak & Schmittgen, 2001). The expression levels of genes were normalized to the 3 housekeeping genes and calibrated to the negative siRNA-treated sample.

## Statistical analyses

Both *in vitro* and *in vivo* data were analyzed using unpaired Student's *t*-test or one-way ANOVA with Dunnett's or Tukey's *post hoc* tests for comparisons between appropriate groups using GraphPad Prism v. 7.0 software. The choroidal and retinal sprouting assays were analyzed using two-way repeated measures ANOVA with Dunnett's *post hoc* tests, while the knockdown time course was analyzed by two-way repeated measures ANOVA with Bonferroni's *post hoc* tests. Two-sided *P*-values < 0.05 were considered significant in all tests.

**Expanded View** for this article is available online.

## Acknowledgements

We thank Michael Sturek for the porcine brain and Harry Dailey for the FECH expression plasmid and protein production advice. This work was supported by grants from the International Retinal Research Foundation, the Retina Research Foundation, the Carl Marshall & Mildred Almen Reeves Foundation, Inc., the Ralph W. and Grace M. Showalter Research Trust Fund, NIH/NCATS KL2TR001106 and UL1TR001108, and NIH/NEI R01EY025641 (T.W.C.); the Ausich Graduate Scholarship from Kemin Health (H.D.B.); R01EY012601, R01EY007739, R01HL110170, R01DK090730, and R01EY025383 (M.B.G.); R01EY025383, R01EY018358, and JDRF 2-SRA-2014-146-Q-R (M.E.B.); the Basic Science Research Program through the National Research Foundation of Korea (NRF) funded by the Ministry of Education (NRF-2013R1A1A2007151) and the Pioneer Research Center Program through the NRF funded by the Ministry of

**The paper explained**

**Problem**
Ocular neovascularization—aberrant new blood vessel growth in the eye—is responsible for much of the vision loss associated with blinding eye diseases such as retinopathy of prematurity, proliferative diabetic retinopathy, and wet age-related macular degeneration. Existing drugs all target vascular endothelial growth factor signaling and are not effective in all patients. We used a chemical proteomic approach to seek novel therapeutic targets for ocular neovascularization.

**Results**
We identified the heme synthesis enzyme ferrochelatase as a target of the antiangiogenic natural product cremastranone. Ferrochelatase knockdown or chemical inhibition blocked proliferation, migration, and tube formation of retinal and choroidal endothelial cells in culture and knockdown or mutation of *Fech* blocked neovascularization in a mouse model. Ferrochelatase was upregulated in human and mouse eyes undergoing neovascularization. The approved drug griseofulvin, which inhibits ferrochelatase as an off-target effect, was an effective oral and intravitreal therapy in a murine choroidal neovascularization model.

**Impact**
Ferrochelatase is a druggable mediator of ocular neovascularization that is a promising target for development of new therapeutic approaches.

Science, ICT & Future Planning (2014M3C1A3001556) (S.-Y.S.); and an unrestricted grant from Research to Prevent Blindness, Inc. (T.W.C., M.B.G., and M.E.B.).

## Author contributions

TWC and HDB conceived the study and drafted the manuscript. HDB performed the *in vitro* experiments with the help of RSS, TS, KLS, CMB, BT, and MS. RSS and XQ performed the *in vivo* experiments with the help of JQ, SA, and KG. RSS and SSPB performed the sprouting assays. S-YS and BL synthesized the affinity reagents. MBG and MEB analyzed data and provided conceptual input. All authors discussed the results and commented on the manuscript.

## Conflict of interest

H.D.B. and T.W.C. are named inventors on a patent application related to this work. The other authors declare no competing financial interests.

## For more information

The Corson laboratory website: http://glick.iu.edu/corson
The *FECH* gene: http://www.ncbi.nlm.nih.gov/gene/2235
BrightFocus Foundation information on AMD: http://www.brightfocus.org/macular

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
