## [Review Process File · EMBO Molecular Medicine]

Ferrochelatase is a therapeutic target for ocular neovascularization

Halesha D. Basavarajappa, Rania S. Sulaiman, Xiaoping Qi, Trupti Shetty, Sardar Sheik Pran Babu, Kamakshi L. Sishtla, Bit Lee, Judith Quigley, Sameerah Alkhairy, Christian M. Briggs, Kamna Gupta, Buyun Tang, Mehdi Shadmand, Maria B. Grant, Michael E. Boulton, Seung-Yong Seo & Timothy W. Corson

Corresponding author: Timothy Corson, Indiana University School of Medicine

Review timeline:

Submission date:	29 April 2016
Editorial Decision:	03 August 2016
Authors Appealed Decision:	03 August 2016
Editorial Decision:	03 August 2017
Revision received:	18 January 2017
Editorial Decision:	15 February 2017
Revision received:	01 March 2017
Accepted:	02 March 2017

Transaction Report:

Editor: Céline Carret

1st Editorial Decision

03 August 2016

Thank you for the submission of your research manuscript to our editorial office. We have now received the enclosed reports on it. As you will see, while referee 3 is rather positive about the study, the other two reviewers, while considering the results potentially interesting, raise serious concerns regarding the conclusiveness of the data and pinpoint several technical issues that preclude a solid interpretation of the experimental evidence provided. All of the reviewers call for a considerable amount of additional experimentation to resolve these issues, both *in vitro* but also, and importantly for our scope, *in vivo* in a more appropriate model.

Given the nature of these criticisms, the amount of work likely to be required to address them, and the fact that EMBO Molecular Medicine can only invite revision of papers that receive enthusiastic support from a majority of referees, I am afraid that we do not feel it would be productive to call for a revised version of your manuscript at this stage and therefore we cannot offer to publish it.

This said, because of the potential interest of the findings, we would have no objection to consider a **new** manuscript on the same topic if at some time in the near future you obtained data that would considerably strengthen the message of the study and address the referees concerns in full. To be completely clear, however, I would like to stress that if you were to send a new manuscript this would be treated as a new submission rather than a revision and would be reviewed afresh, in

particular with respect to the literature and the novelty of your findings at the time of resubmission. If you decide to follow this route, please make sure you nevertheless upload a letter of response to the referees' comments.

At this stage of analysis, though, I am sorry to have to disappoint you. I nevertheless hope, that the referee comments will be helpful in your continued work in this area and I thank you for considering EMBO Molecular Medicine.

***** Reviewer's comments *****

Referee #1 (Comments on Novelty/Model System):

The work described in the manuscript is interesting and presents ferrochelatase (FECH) as a possible new druggable target to prevent neovascularization in wet age-related macular degeneration. Oral delivery of the drug appears to be as effective, at least in the same order of magnitude, as intravitreal injection in the laser-induced choroidal neovascularization (L-CNV) mouse model that was used. One issue that seemed to stand out however, is that although FECH expression was seen throughout the retinal GCL and ONL layers of human retinas (Fig 3), the L-CNV mouse model only addresses the inhibition of FECH expression by intravitreal injection of FECH siRNA in the choroidal layer. It is not evident how this would translate into clinical applications to treat human AMD, given the known anatomical differences between the retinal layers in humans and mice. Also in the same figure, it's not clear how graphs in Fig 3C and Fig 3E are directly comparable since the graph in Fig 3C appears to be quantified from images showed in Fig 3B (cross section of the human retina) vs. the graph in Fig 3E is quantified from images shown in Fig. 3D (whole mount staining of RPE/choroid from L-CNV eyes).

Referee #1 (Remarks):

- An initial paragraph that summarizes the major findings and the referee's overall impressions, as well as highlighting major shortcomings of the manuscript.
- Specific numbered comments, which may be broken down into major and minor criticisms if appropriate (numbering facilitates both the editor's evaluation of the manuscript and the authors' rebuttal to the report).

Overall Review

The studies described in the manuscript entitled, "Ferrochelatase is a therapeutic target for ocular neovascularization", presents a detailed study in which the authors use a "forward chemical genetic approach" to find protein targets for cremastranone, a known anti-angiogenic natural product. The authors used photoaffinity chromatography to isolate cremastranone-bound proteins from tissue lysates of porcine brain (reason unknown). From this work, they identified 2 proteins, ferrochelatase (FECH) as well as another protein, which is not discussed further. FECH is the terminal enzyme in heme biosynthesis and catalyzes the insertion of iron into protoporphyrin IX. Overexpression of FECH was found in human age-related macular degeneration (AMD) eyes. In vivo studies using a mouse laser-induced choroidal neovascularization model (L-CNV) showed FECH knockdown or its inhibition with the oral anti-fungal drug, griseofulvin, reduced choroidal neovascularization and modulates endothelial nitric oxide function and VEGFR2 expression. The authors concluded from these studies that FECH is a druggable mediator of angiogenesis whose inhibition could be used therapeutically to effectively block ocular neovascularization. A model was proposed that shows how the production of heme by FECH in the mitochondria affects eNOS, HIF1 α and VEGFR2 levels in the cytoplasm, which eventually impacts angiogenesis. Overall, the studies follow a logical progression to understand how the inhibition of FECH impacts HRECs by the use of siRNA and FECH inhibition by N-methyl protoporphyrin (NMPP) and griseofulvin. After this analysis however, the authors show very general schematic of how FECH affects angiogenesis which detracted from their careful study of FECH inhibition by griseofulvin and NMPP via P450 involvement, in vivo observations in the L-CNV mice, and downstream effects on VEGFR2, eNOS and HIF-1 α .

Interestingly, inhibition of FECH by NMPP results in decreased proliferation, migration and tube formation and is specific for retinal endothelial cells but does not cause apoptosis. The authors

conclude that NMPP inhibition of HRECs is cytostatic not cytotoxic, but this conclusion seems premature since they do not provide a time course at any dose (Fig 2F). Although FECH expression is not limited to a specific layer in the human AMD retina (Fig 3B), it's not clear what the distribution of FECH expression is in the whole mount staining of the mouse L-CNV retina (Fig 3D). Also, if the Y-axis in the graph in Fig 3C reads, "FECH signal (% of Normal), i.e., "normalized", shouldn't the "Normal" bar be "0" instead of 100 + 25? The results in Fig 5 are particularly compelling, and the authors should note that the effects of the oral or intravitreal administration of griseofulvin is in the same order of magnitude. VEGFR2 levels were rescued by the addition of Hemin (Fig 6E) however, the level of eNOS was not rescued the presence of Hemin as stated. In fact, eNOS increased only by 20% compared to the measured eNOS level in the presence of FECH siRNA, but was still 35% lower than the eNOS levels attained in the presence of the negative siRNA control. Fortunately, since nitric oxide stabilizes HIF-1 α , lower levels of eNOS results in decreased levels of nitric oxide, decreased HIF-1 α levels, and presumably, decreased angiogenesis. Did the effect of FECH inhibition affect expression of other VEGFRs or neuropilin-1? Perhaps combination of griseofulvin with anti-VEGF biologics may have a synergistic effect and possibly decrease side effects.

Specific Points

- 1) In the Introduction, Page 3, line 37, add the word "as" in the sentence, "...vision loss as a direct.."
- 2) In the Results, Page 8 lines 166-167, need to rewrite this since VEGFR2 was rescued but not the level of eNOS (see the explanation in the 2nd paragraph of the Overall Review).
- 3) In the Results, Page 8, line 169, "The decreased protein levels..." is vague, specify which proteins
- 4) In the Materials and Methods, Page 13, lines 289 and 293, the use of the word "dounced" as a verb is incorrect. The supernatant was homogenized using a Dounce homogenizer.
- 5) In the Materials and Methods, Lines 341, 352, 362, 442, 447, 449, 454, 472, 484 and 493, remove the word "then".
- 6) In the Materials and Methods, Page 16, line 352, the word "compound" is vague. Is it griseofulvin or NMPP or both?
- 7) In the Materials and Methods, Page 18, line 391, add the word "the" at the end of the line
- 8) In the Materials and Methods, Page 18, lines 298 and 299, what are the details of this labeling, concentration and conditions used?
- 9) In the Materials and Methods, Page 18, line 406, add the word "humor" at the end of the line
- 10) In the Materials and Methods, Page 18, line 412, Delete the word, "tissues" and substitute "sections or whole mounts".
- 11) In the Materials and Methods, Page 18, line 415, add the word "the" between the words "...label retinal..."
- 12) In the Materials and Methods, Page 19, line 422, delete the word "euthanized".
- 13) In the Materials and Methods, Page 19, line 425, add the word, "the" to the end of the line
- 14) In the Materials and Methods, Page 19, line 437, add the word "described" between the words, "...as above".
- 15) In the Materials and Methods, Page 20, line 445, add the manufacturer for the Synergy Plate reader.
- 16) In the Materials and Methods, Page 20, line 447, substitute the word "confluency" for "confluent".
- 17) In the Materials and Methods, Page 20, line 454, the S in the word "supernatant..." should be capitalized.
- 18) In the Materials and Methods, Page 21, line 466, replace the phrase, "The eluates then analyzed in 4-... with "Eluates were separated in a 4-..."
- 19) In the Materials and Methods, Page 21, line 468, replace the word "and" with "or" and "Tubulin" with "tubulin"
- 20) In the Materials and Methods, Page 21, line 469, delete "of the" and "...Buffered Saline" should be "...buffered saline".
- 21) In the Materials and Methods, Page 21, lines 473 and 483, the word "compounds or compound" is vague, can you list which compound(s) you are using here?
- 22) In the Materials and Methods, Page 21, line 474, add the word "the" between the words "...by addition..."
- 23) In the Materials and Methods, Page 21, line 486, replace "...represents iron" with "represented the degree of iron..."
- 24) In the Materials and Methods, Page 23, line 513, please reference or explain what you did using

the " $\Delta\Delta C1$ method".

25) In the Materials and Methods, Page 23, line 514, what do you mean by "calibrated to"? Is this "normalized to"?

26) In the Materials and Methods, Page 23, line 518, replace "...groups as appropriate in" with "...appropriate groups using..."

27) In the References, Page 26, lines 610, 611, Delete this reference, cannot reference a manuscript that isn't published.

Referee #2 (Comments on Novelty/Model System):

This paper is quite simple and at certain level quite naïve in the way angiogenesis is tested in vitro, lacking definitive in vivo proof to place FECH in in vivo angiogenesis (e.g. model systems). I suggest major revision with a comparison with different EC subtypes (retina, microvascular, macrovascular) for the vitro assays and genetic model in vivo to support their hypothesis.

Referee #2 (Remarks):

By using a chemical approach Basavarajappa H.D. et al identified the heme synthesis enzyme ferrochelatase (FECH) as a key player in neovascularization occurring during age-related macular degeneration. They show that ferrochelatase inhibition can indeed lead to inhibition of pathological angiogenesis through the inhibition of VEGFR2/eNOS/Hif1alpha signalling pathway. Interestingly they propose an FDA-approved anti-fungal drug griseofulvin as a therapeutic option to inhibit ferrochelatase and eventually cure AMD.

Although an interesting topic, the data and experimental settings used in the paper could be more detailed and technically sound. The study nicely fit within the scope of EMM since it offers a putative link among basic biology and clinical research. However, such work should propose studies based on model organism to fully fall within the scope of the journal. Therefore, to fully prove the role of FECH a genetic model of FECH KO should be employed.

Some points have to be addressed, as follows:

Major points:

- 1) The data shown here point to the role of ferrochelatase during angiogenesis as a key enzyme in a VEGFR2-dependent signalling pathway. Furthermore, they never show that inhibition of ferrochelatase can be a useful approach to treat vegf-resistant or refractory AMD. Recently it has been shown that many growth factors other than VEGF may mediate ocular neovascularization, indicating that multi target approaches are more promising than single target strategies. The authors should comment on this and provide, if possible, any evidence that inhibition of ferrochelatase can improve the available treatment for ocular angiogenesis-dependent pathologies.
- 2) The authors use human retinal endothelial cells (HREC) for their in vitro experiments. Also they use the choroid ex-vivo assay to assess choroidal angiogenesis. Since different angiogenesis-dependent ocular pathologies affect different vascular districts (namely the retinal vessels in proliferative diabetic retinopathy and choroidal vessels in wet age-related macular degeneration) and given the specificity of ferrochelatase inhibition observed when comparing HUVEC vs HREC, the authors should provide more experimental evidences to sustain the hypothesis that ferrochelatase is a druggable target which is specific for the ocular vasculature. The authors should repeat the key experiments shown in fig. 2, 4 and 6 by using choroidal cells (either primary or Rf/6a cell line) as well as include in fig. 4 an ex-vivo murine retina angiogenesis assay. This would help to prove that ferrochelatase is a specific and universal druggable target for all ocular angiogenesis-dependent pathologies. Otherwise, if their focus is AMD, I would suggest to use choroidal endothelial cells instead of HREC.
- 3) The authors propose griseofulvin as a potential therapeutic to inhibit ferrochelatase activity and eventually block pathological angiogenesis during AMD. Nevertheless they do not provide evidence that griseofulvin acts via ferrochelatase inhibition instead of via a microtubule-mediated block of mitosis. Does hemin addition to the cell culture medium rescue the block in proliferation they observe? (fig. 4A and 4D). Does 100 μ M griseofulvin give cytotoxic effects? The author should also test the effect of microtubule inhibitor devoid of anti-ferrochelatase activity as control. Can the authors quantify the amount of active griseofulvin in the blood stream of mice feed ad libitum with

this antifungal drug ?

4) It could be nice if authors can show in fig.6 some in vivo staining to validate on human eye sections or murine choroid tissue the molecular mechanism they propose in vitro.

5) In Fig 6, the experiments have been performed without VEGF administration. Is then VEGFR activation ligand-independent in these cells and conditions ? Isn't it expected to have reduced eNOS activity if the total eNOS level is low ? what the correlation here ?

Referee #3 (Remarks):

In their manuscript entitled "Ferrochelatase is a therapeutic target for ocular neovascularization", Basavarajappa et. al. describe the identification of the molecular target of an antiangiogenic compound as ferrochelatase. They then proceed to show that ferrochelatase is necessary for proliferation and migration of retinal endothelial cells and is overexpressed in choroidal neovascularization lesions in a mouse model and human AMD patients. Notwithstanding, they show that ferrochelatase can be inhibited by an FDA-approved compound to decrease neovascularization in a mouse model. Lastly, they probe the mechanism of action of ferrochelatase in angiogenesis and show that it involves eNOS, HIF-1a and VEGFR2.

This manuscript is of excellent quality and we believe it should be accepted. We have a few comments that may lead to its further improvement.

1) In supplementary figure 4 the authors show that FECH knockdown by siRNA does not affect the proliferation of ARPE19, 92-1 or HUVEC cells. Toxicity in these cell lines should also be assessed for the chemical inhibitors of FECH.

2) The compound used to purify the cremastranone-binding molecules in the photoaffinity chromatography is not identical to cremastranone. More specifically, compound (3) in Figure 1 appears to have two of cremastranone's hydrogens substituted for methyl groups. The authors should mention why that needed to be done.

3) In figure 3B, the human eye sections are not of great quality. Specifically, the "normal" eye section has a disorganized GCL while the AMD and no-primary eye sections are almost completely missing their GCL. In addition, the ONL for all sections is surprisingly thin. At 2-3 cell nuclei and 20 um of thickness the ONLs appear abnormal - at least for the control eyes.

4) In supplementary figure 3 D and E, is the % of apoptotic cells significantly different between the highest doses of the drugs (25 um for NMPP and 50 um for Griseofulvin) and the control treatment? If so, the authors should state that. In addition, the sentence "FECH knockdown and low-dose chemical inhibition were not associated with increased apoptosis for these cells" should be modified to reflect the minimal -yet existent - apoptosis that was observed with the inhibitor treatments.

5) The n for each experiment/experimental group, as well as the specific statistical test performed should be mentioned at the respective figure subcaption. Currently, the authors describe the n as "n {greater than or equal to}3" and the statistical test as "Student's t-test or ANOVA with..." at the end of the caption.

Authors appeal decision

03 August 2016

Thank you for considering our manuscript, referenced above. We greatly appreciate the reviewers' thorough and constructive commentary, and their general enthusiasm for the novelty and importance of our study.

Although we recognize that substantial work is required for a revision, we believe that this can be accomplished in a timely fashion. In particular, during the review period, we obtained and established a colony of Fech knockout mice, so the further in vivo experiments requested by reviewer #2 can be accomplished quite quickly. We have also already completed some of the requested in vitro experiments.

2nd Editorial Decision

03 August 2017

I have looked again at your manuscript and discussed it within our team. In light of your arguments and the availability of FECH-KO mice, we would like to give you a chance to address all issues

raised by all three referees, experimentally when required. Please bear in mind that revising now will not ascertain publication later on and your revision will be reviewed once more from the same set of referees.

Please proceed to revising your article according to our guidelines (see below).

Revised manuscripts should be submitted within three months of a request for revision; they will otherwise be treated as new submissions, except under exceptional circumstances in which a short extension is obtained from the editor.

I look forward to seeing a revised form of your manuscript as soon as possible.

2nd Revision - authors' response

18 January 2017

Referee #1

The work described in the manuscript is interesting and presents ferrochelatase (FECH) as a possible new druggable target to prevent neovascularization in wet age-related macular degeneration. Oral delivery of the drug appears to be as effective, at least in the same order of magnitude, as intravitreal injection in the laser-induced choroidal neovascularization (L-CNV) mouse model that was used. One issue that seemed to stand out however, is that although FECH expression was seen throughout the retinal GCL and ONL layers of human retinas (Fig 3), the L-CNV mouse model only addresses the inhibition of FECH expression by intravitreal injection of FECH siRNA in the choroidal layer. It is not evident how this would translate into clinical applications to treat human AMD, given the known anatomical differences between the retinal layers in humans and mice.

We thank the reviewer for this favorable assessment. To further validate our findings and increase translational relevance, in this revision we present new data showing that FECH inhibition in primate choroidal endothelial cells has the predicted antiangiogenic effect (Fig. EV 3), as does FECH inhibition in murine retina ex vivo (Fig. 4E). Importantly, we have also shown that Fech mutation in a genetic mouse model decreases L-CNV (Figure 3E).

Also in the same figure, it's not clear how graphs in Fig 3C and Fig 3E are directly comparable since the graph in Fig 3C appears to be quantified from images showed in Fig 3B (cross section of the human retina) vs. the graph in Fig 3E is quantified from images shown in Fig. 3D (whole mount staining of RPE/choroid from L-CNV eyes).

To make the human and mouse staining more directly comparable, we have added cross section staining of murine L-CNV to this figure (new Fig. 3B), indicating that Fech is regulated through all layers of the mouse eye undergoing neovascularization, but notably in the neovessels.

The studies described in the manuscript entitled, "Ferrochelatase is a therapeutic target for ocular neovascularization", presents a detailed study in which the authors use a "forward chemical genetic approach" to find protein targets for cremastranone, a known anti-angiogenic natural product. The authors used photoaffinity chromatography to isolate cremastranone-bound proteins from tissue lysates of porcine brain (reason unknown).

We used porcine brain for the pulldown experiments as a low-risk and readily available rich source of protein. This is now noted in the manuscript, l. 319. Preliminary experiments with bovine eye tissue failed to provide enough input protein for successful pulldowns.

From this work, they identified 2 proteins, ferrochelatase (FECH) as well as another protein, which is not discussed further. FECH is the terminal enzyme in heme biosynthesis and catalyzes the insertion of iron into protoporphyrin IX. Overexpression of FECH was found in human age-related macular degeneration (AMD) eyes. In vivo studies using a mouse laser-induced choroidal neovascularization model (L-CNV) showed FECH knockdown or its inhibition with the oral anti-fungal drug, griseofulvin, reduced choroidal neovascularization

and modulates endothelial nitric oxide function and VEGFR2 expression. The authors concluded from these studies that FECH is a druggable mediator of angiogenesis whose inhibition could be used therapeutically to effectively block ocular neovascularization. A model was proposed that shows how the production of heme by FECH in the mitochondria affects eNOS, HIF1 α and VEGFR2 levels in the cytoplasm, which eventually impacts angiogenesis. Overall, the studies follow a logical progression to understand how the inhibition of FECH impacts HRECs by the use of siRNA and FECH inhibition by N-methyl protoporphyrin (NMPP) and griseofulvin.

We thank the reviewer for these favorable comments.

After this analysis however, the authors show very general schematic of how FECH affects angiogenesis, which detracted from their careful study of FECH inhibition by griseofulvin and NMPP via P450 involvement, in vivo observations in the L-CNV mice, and downstream effects on VEGFR2, eNOS and HIF-1 α .

In response to this suggestion, in this revision, we have deleted the general model schematic (previous Fig. 6G).

Interestingly, inhibition of FECH by NMPP results in decreased proliferation, migration and tube formation and is specific for retinal endothelial cells but does not cause apoptosis. The authors conclude that NMPP inhibition of HRECs is cytostatic not cytotoxic, but this conclusion seems premature since they do not provide a time course at any dose (Fig 2F).

We have added the results of a washout experiment for NMPP and griseofulvin's effects on HRECs (Fig. EV 2E, G), indicating that cells are still viable and bounce back after these compounds are removed, strongly supporting a cytostatic effect.

Although FECH expression is not limited to a specific layer in the human AMD retina (Fig 3B), it's not clear what the distribution of FECH expression is in the whole mount staining of the mouse L-CNV retina (Fig 3D).

We have added staining for Fech in sections of mouse L-CNV eyes (Fig. 3B) to clarify this distribution; as in humans, Fech was expressed throughout the retina.

Also, if the Y-axis in the graph in Fig 3C reads, "FECH signal (% of Normal), i.e., "normalized", shouldn't the "Normal" bar be "0" instead of 100 + 25?

The displayed data in Fig. 3C are expressed as a percentage of the normal signal, not % over normal, so a value of 100% for the normal average is correct as shown.

The results in Fig 5 are particularly compelling, and the authors should note that the effects of the oral or intravitreal administration of griseofulvin is in the same order of magnitude.

We have added this important note to the text (l. 165).

VEGFR2 levels were rescued by the addition of Hemin (Fig 6E) however, the level of eNOS was not rescued the presence of Hemin as stated. In fact, eNOS increased only by 20% compared to the measured eNOS level in the presence of FECH siRNA, but was still 35% lower than the eNOS levels attained in the presence of the negative siRNA control.

We agree that hemin does not completely rescue eNOS levels after FECH knockdown, and have noted that the rescue was "partial" (l. 189). However, although eNOS levels after hemin treatment remained lower than in control, they were statistically significantly higher than in *FECH* siRNA-treated cells, as now clarified in Fig. 6E and its legend.

Fortunately, since nitric oxide stabilizes HIF-1 α , lower levels of eNOS results in decreased levels of nitric oxide, decreased HIF-1 α levels, and presumably, decreased angiogenesis. Did the effect of FECH inhibition affect expression of other VEGFRs or neuropilin-1?

We have investigated this question, and found that *FECH* knockdown did not change the protein levels of neuropilin or VEGFR1, as now indicated in Fig. 6E.

Perhaps combination of griseofulvin with anti-VEGF biologics may have a synergistic effect and possibly decrease side effects.

To address this point, we assessed the combination of griseofulvin with anti-VEGF₁₆₄ antibody in the L-CNV model (Fig. EV 5C). The addition of a low dose of anti-VEGF₁₆₄ to griseofulvin treatment caused a slight reduction in CNV lesion volume compared to griseofulvin alone, but this difference was not significant. Further investigation of the combination of griseofulvin with anti-VEGF therapy is warranted.

Specific Points

1) In the Introduction, Page 3, line 37, add the word "as" in the sentence, "..vision loss as a direct.."

Corrected, l. 39.

2) In the Results, Page 8 lines 166-167, need to rewrite this since VEGFR2 was rescued but not the level of eNOS (see the explanation in the 2nd paragraph of the Overall Review).

Revised – see above, l. 189.

3) In the Results, Page 8, line 169, "The decreased protein levels..." is vague, specify which proteins

Corrected, l. 191.

4) In the Materials and Methods, Page 13, lines 289 and 293, the use of the word "dounced" as a verb is incorrect. The supernatant was homogenized using a Dounce homogenizer.

Corrected, ll. 321 and 325-326.

5) In the Materials and Methods, Lines 341, 352, 362, 442, 447, 449, 454, 472, 484 and 493, remove the word "then".

Corrected, ll. 367, 374, 386, 390, 397, 499, 505, 506, 511, 520, 529, 551.

6) In the Materials and Methods, Page 16, line 352, the word "compound" is vague. Is it griseofulvin or NMPP or both?

Corrected to indicate that both compounds were added by this method, l. 386.

7) In the Materials and Methods, Page 18, line 391, add the word "the" at the end of the line

Corrected, l. 432.

8) In the Materials and Methods, Page 18, lines 298 and 299, what are the details of this labeling, concentration and conditions used?

To further clarify the UV labeling, we have added the measured irradiance output of the UV lamp (l. 332).

9) In the Materials and Methods, Page 18, line 406, add the word "humor" at the end of the line

Corrected, l. 461.

10) In the Materials and Methods, Page 18, line 412, Delete the word, "tissues" and substitute "sections or whole mounts".

Corrected, l. 447.

11) In the Materials and Methods, Page 18, line 415, add the word "the" between the words "...label retinal..."

Corrected, l. 457.

12) In the Materials and Methods, Page 19, line 422, delete the word "euthanized".

Corrected, l. 465.

13) In the Materials and Methods, Page 19, line 425, add the word, "the" to the end of the line

Corrected, l. 468.

14) In the Materials and Methods, Page 19, line 437, add the word "described" between the words, "...as above".

Corrected, l. 494.

15) In the Materials and Methods, Page 20, line 445, add the manufacturer for the Synergy Plate reader.

This is mentioned previously, but added again here for clarity, l. 502.

16) In the Materials and Methods, Page 20, line 447, substitute the word "confluency" for "confluent".

Corrected, l. 504.

17) In the Materials and Methods, Page 20, line 454, the S in the word "supernatant..." should be capitalized.

Corrected, l. 511.

18) In the Materials and Methods, Page 21, line 466, replace the phrase, "The eluates then analyzed in 4-... with "Eluates were separated in a 4-..."

Corrected, ll. 523-524.

19) In the Materials and Methods, Page 21, line 468, replace the word "and" with "or" and "Tubulin" with "tubulin"

Corrected, ll. 525-526.

20) In the Materials and Methods, Page 21, line 469, delete "of the" and "...Buffered Saline" should be "...buffered saline".

Corrected, ll. 526-527.

21) In the Materials and Methods, Page 21, lines 473 and 483, the word "compounds or compound" is vague, can you list which compound(s) you are using here?

Corrected, ll. 530 and 541.

22) In the Materials and Methods, Page 21, line 474, add the word "the" between the words "...by addition..."

Corrected, l. 531.

23) In the Materials and Methods, Page 21, line 486, replace "...represents iron" with "represented the degree of iron..."

Corrected, l. 545.

24) In the Materials and Methods, Page 23, line 513, please reference or explain what you did using the " $\Delta\Delta C_1$ method".

We have added a reference (Livak and Schmittgen, 2001) on l. 572, which describes this standard comparative method for quantifying qPCR data.

25) In the Materials and Methods, Page 23, line 514, what do you mean by "calibrated to"? Is this "normalized to"?

In the $\Delta\Delta C_t$ method, threshold cycle (C_t) numbers for gene(s) of interest are first normalized to housekeeping gene(s) C_t s, then "calibrated" (compared) to the normalized values for a single control sample. Thus reported values are expressed as a quantity relative to the calibrator sample. Please see Livak and Schmittgen as now referenced in the text (l. 572) for details.

26) In the Materials and Methods, Page 23, line 518, replace "...groups as appropriate in" with "...appropriate groups using..."

Corrected, l. 576.

27) In the References, Page 26, lines 610, 611, Delete this reference, cannot reference a manuscript that isn't published.

This manuscript is now published and the reference has been updated to reflect this (ll. 696-698).

Referee #2 (Comments on Novelty/Model System):

This paper is quite simple and at certain level quiet naïve in the way angiogenesis is tested in vitro, lacking definitive in vivo proof to place FECH in in vivo angiogenesis (e.g. model systems). I suggest major revision with a comparison with different EC subtypes (retina, microvascular, macrovascular) for the vitro assays and genetic model in vivo to support their hypothesis.

In this revision, we have added further analysis of the effects of FECH knockdown and/or inhibition in other cell types in vitro. Antiproliferative effects were remarkably selective for microvascular cells (HRECs, a choroidal endothelial cell line Rf/6a, and brain microvascular endothelial cells) versus other ocular cell lines and HUVECs (Figs. 2, EV 3 and EV 4). In addition, we have assessed L-CNV in a *Fech* mutant mouse in vivo and observed a reduction of neovascularization in the mutant (see below for details), further supporting our hypothesis.

Referee #2 (Remarks):

By using a chemical approach Basavarajappa H.D. et al identified the heme synthesis enzyme ferrochelatase (FECH) as a key player in neovascularization occurring during age-related macular degeneration. They show that ferrochelatase inhibition can indeed lead to inhibition of pathological angiogenesis through the inhibition of VEGFR2/eNOS/Hif1alpha signalling pathway. Interestingly they propose an FDA-approved anti-fungal drug griseofulvin as a therapeutic option to inhibit ferrochelatase and eventually cure AMD.

Although an interesting topic, the data and experimental settings used in the paper could be more detailed and technically sound. The study nicely fit within the scope of EMM since it offers a putative link among basic biology and clinical research. However, such work should propose studies based on model organism to fully fall within the scope of the journal. Therefore, to fully prove the role of FECH a genetic model of FECH KO should be employed.

In response to this excellent suggestion, we have assessed L-CNV in *Fech* mutant mice. Excitingly, neovascularization was reduced in heterozygous partial loss-of-function *Fech* mutants compared to wild-type, and further reduced in homozygous mutants (Fig. 3E). We feel that this new finding provides compelling evidence of the importance of *Fech* for neovascularization.

Some points have to be addressed, as follows:

Major points:

1) The data shown here point to the role of ferrochelatase during angiogenesis as a key enzyme in a VEGFR2-dependent signalling pathway. Furthermore, they never show that inhibition of ferrochelatase can be a useful approach to treat vegf-resistant or refractory AMD. Recently it has been shown that many growth factors other than VEGF may mediate ocular neovascularization, indicating that multi target approaches are more promising than single target strategies. The authors should comment on this and provide, if possible, any evidence that inhibition of ferrochelatase can improve the available treatment for ocular angiogenesis-dependent pathologies.

This is an excellent point. To our knowledge, there are no model systems that adequately reflect VEGF-resistant or refractory wet AMD. However, we agree that multi-targeted, combination therapies are an appealing way to potentially avoid a loss of response in the clinic. Towards this end, we now include data on the combination of griseofulvin with an anti-VEGF antibody in the L-CNV model. The addition of a low dose of anti-VEGF to griseofulvin treatment caused a slight reduction in CNV lesion volume compared to griseofulvin alone, but this difference was not significant (Fig. EV 5C). Further investigation of the combination of griseofulvin with anti-VEGF is warranted.

2) The authors use human retinal endothelial cells (HREC) for their in vitro experiments. Also they use the choroid ex-vivo assay to assess choroidal angiogenesis. Since different angiogenesis-dependent ocular pathologies affect different vascular districts (namely the retinal vessels in proliferative diabetic retinopathy and choroidal vessels in wet age-related macular degeneration) and given the specificity of ferrochelatase inhibition observed when comparing HUVEC vs HREC, the authors should provide more experimental evidences to sustain the hypothesis that ferrochelatase is a druggable target which is specific for the ocular vasculature. The authors should repeat the key experiments shown in fig. 2, 4 and 6 by using choroidal cells (either primary or Rf/6a cell line) as well as include in fig. 4 an ex-vivo murine retina angiogenesis assay. This would help to prove that ferrochelatase is a specific and universal druggable target for all ocular angiogenesis-dependent pathologies. Otherwise, if their focus is AMD, I would suggest to use choroidal endothelial cells instead of HREC.

As the reviewer suggests, to increase relevance, we have repeated key experiments in Rf/6a cells, a primate choroidal endothelial cell line. Like HRECs, these cells were sensitive to NMPP and griseofulvin (Fig. EV 3) in a way that other cell types were not (Fig. EV 4). We have also now tested griseofulvin in an ex vivo retinal angiogenesis assay as suggested (Fig. 4E), where it showed similar effects to those we observed in the choroidal sprouting assay.

3) The authors propose griseofulvin as a potential therapeutic to inhibit ferrochelatase activity and eventually block pathological angiogenesis during AMD. Nevertheless they do not provide evidence that griseofulvin acts via ferrochelatase inhibition instead of via a microtubule-mediated block of mitosis. Does hemin addition to the cell culture medium rescue the block in proliferation they observe? (fig. 4A and 4D).

Unfortunately, since hemin itself can cause toxicity, assessing its rescuing effects on proliferation subsequent to griseofulvin treatment is challenging. However, the lack of apoptosis observed in cells treated with antiangiogenic concentrations of griseofulvin argues against a mitosis blockade, as the usual cellular response to a sustained mitosis block is apoptosis. The reversibility of griseofulvin's antiproliferative effects on HRECs (Fig. EV 2), plus its limited antiproliferative effects on other cell types (Fig. EV 4) also argue strongly against a mitosis block as mechanism.

Does 100µM griseofulvin give cytotoxic effects ?

In the ex vivo retinal angiogenesis assay, continuous exposure to 100 µM griseofulvin did lead to toxicity and loss of existing sprouts (data not shown). However, this is not directly comparable to

the in vivo conditions in Fig. 5, as in the mouse eye, 100 μ M griseofulvin was the estimated initial concentration in the vitreous after a single injection, which would be quickly diluted throughout the eye. The appearance of the treated retina on OCT imaging (Fig. 5D) one week after injection argues against any toxic effects in this system, as does the aforementioned lack of in vitro toxicity on other ocular cell types (Fig. EV 4) and the lack of ocular side effects in humans undergoing systemic griseofulvin treatment.

The author should also test the effect of microtubule inhibitor devoid of anti-ferrochelatase activity as control.

Unfortunately, such an experiment would be difficult to interpret, as microtubule inhibitors are inherently antiproliferative, or even cytotoxic, so would be expected to have a (non-specific) anti-angiogenic effect as well. The lack of antiproliferative effects of griseofulvin on non-microvascular cell types (Fig. EV 4) argues that it is not acting as a microtubule inhibitor at effective antiangiogenic concentrations.

Can the authors quantify the amount of active griseofulvin in the blood stream of mice fed ad libidum with this antifungal drug ?

We attempted quantification of griseofulvin in blood of mice fed griseofulvin. The clinical HPLC assay used was not able to provide accurate quantification, but qualitatively, griseofulvin was present in the treated animals, and absent in the untreated. However, more importantly, we were able to quantify NMPP and PPIX in the livers of treated mice, indicating metabolism of griseofulvin and inhibition of FECH, respectively. PPIX levels were 3.7 ± 1.5 , 72 ± 30 , and 163 ± 50 μ g/g in the 0, 0.5, and 1.0% griseofulvin groups, respectively, while NMPP levels were 0, 680 ± 217 , and 611 ± 127 ng/g. Mean \pm SEM, n = 4. These findings confirm that NMPP is produced after griseofulvin feeding.

4) It could be nice if authors can show in fig.6 some in vivo staining to validate on human eye sections or murine choroid tissue the molecular mechanism they propose in vitro.

While we agree that this would be valuable, such experiments would be complex and may not yield useful data due to the variabilities of timing of the signaling changes during/after treatment. This will be an interesting topic for future work.

5) In Fig 6, the experiments have been performed without VEGF administration. Is then VEGFR activation ligand-independent in these cells and conditions ?

These experiments were performed with VEGF present in the context of complete endothelial growth medium 2. We have now noted this in the Fig. 6 legend, l. 879. We do not expect HRECs to have VEGF-independent VEGFR activation.

Isn't it expected to have reduced eNOS activity if the total eNOS level is low ? what the correlation here ?

Yes, reduced eNOS protein is expected to lead to reduced NOS activity. We include NOS activity data in the manuscript (Fig. 6B) to confirm that FECH knockdown is associated with a functional loss of NOS activity, as well as decreased eNOS protein.

Referee #3 (Remarks):

In their manuscript entitled "Ferrochelatase is a therapeutic target for ocular neovascularization", Basavarajappa et. al. describe the identification of the molecular target of an antiangiogenic compound as ferrochelatase. They then proceed to show that ferrochelatase is necessary for proliferation and migration of retinal endothelial cells and is overexpressed in choroidal neovascularization lesions in a mouse model and human AMD patients. Notwithstanding, they show that ferrochelatase can be inhibited by an FDA-approved compound to decrease neovascularization in a mouse model. Lastly, they probe the mechanism of action of ferrochelatase in angiogenesis and show that it involves eNOS, HIF-1 α and VEGFR2.

This manuscript is of excellent quality and we believe it should be accepted. We have a few comments that may lead to its further improvement.

We thank the reviewer for this favorable assessment.

1) In supplementary figure 4 the authors show that FECH knockdown by siRNA does not affect the proliferation of ARPE19, 92-1 or HUVEC cells. Toxicity in these cell lines should also be assessed for the chemical inhibitors of FECH.

We have tested this, and show negligible antiproliferative effects of NMPP and griseofulvin on these cell types (Fig. EV 4).

2) The compound used to purify the cremastranone-binding molecules in the photoaffinity chromatography is not identical to cremastranone. More specifically, compound (3) in Figure 1 appears to have two of cremastranone's hydrogens substituted for methyl groups. The authors should mention why that needed to be done.

In the Methods (ll. 304-307), we have now explained that this trimethoxy cremastranone analog was incorporated into affinity reagents due to synthetic ease, but that this analog retains activity and selectivity, as reported (Basavarajappa et al, 2015, Lee et al, 2016).

3) In figure 3B, the human eye sections are not of great quality. Specifically, the "normal" eye section has a disorganized GCL while the AMD and no-primary eye sections are almost completely missing their GCL. In addition, the ONL for all sections is surprisingly thin. At 2-3 cell nuclei and 20 um of thickness the ONLs appear abnormal - at least for the control eyes.

We apologize for the quality of these imaged sections, which were poorly chosen; a nearby blood vessel (not visible in the image) displaced the GCL in the "normal" section. We have replaced these images in new Fig. 3C. The ONL remains thin in these sections, but in our experience this is not uncommon in eyes harvested from elderly donors.

4) In supplementary figure 3 D and E, is the % of apoptotic cells significantly different between the highest doses of the drugs (25 um for NMPP and 50 um for Griseofulvin) and the control treatment? If so, the authors should state that.

The % apoptotic cells does not differ significantly from the control, even at the highest compound doses. We have clarified this in the revised Fig. EV 2D, E.

In addition, the sentence "FECH knockdown and low-dose chemical inhibition were not associated with increased apoptosis for these cells" should be modified to reflect the minimal - yet existent - apoptosis that was observed with the inhibitor treatments.

Corrected as suggested, l. 101.

5) The n for each experiment/experimental group, as well as the specific statistical test performed should be mentioned at the respective figure subcaption. Currently, the authors describe the n as "n {greater than or equal to}3" and the statistical test as "Student's t-test or ANOVA with..." at the end of the caption.

We have corrected this as suggested for all figure legends (pp. 32-42), which are now formatted according to *EMBO Mol Med* guidelines.

In this revision, we have also included "For more information" and "The Paper Explained" sections (p. 27) to the main manuscript file, and we have attached as separate files a synopsis, a synopsis image, and an author checklist. We here upload the revised manuscript with changes tracked, a clean version of the revised manuscript, six Figures, five Expanded View figures, and an Appendix containing two Supplementary Figures. Please also note that we have added authors to the manuscript who performed the revision experiments. All authors have agreed to this revised author list. Thank you again for considering our revised manuscript.

3rd Editorial Decision

15 February 2017

Thank you for the submission of your revised manuscript to EMBO Molecular Medicine. We have now received the enclosed reports from the referees that were asked to re-assess it. As you will see the reviewers are now supportive and I am pleased to inform you that we will be able to accept your manuscript pending the following final amendments:

1) Please address the minor changes commented by referee 1. Please provide a letter INCLUDING the reviewer's reports and your detailed responses to their comments (as Word file).

Please submit your revised manuscript within two weeks. I look forward to seeing a revised form of your manuscript as soon as possible.

***** Reviewer's comments *****

Referee #1 (Comments on Novelty/Model System):

The work described in the manuscript is interesting and presents ferrochelatase (FECH) as a possible new druggable target to prevent neovascularization in wet age-related macular degeneration. Oral delivery of the drug appears to be as effective, at least in the same order of magnitude, as intravitreal injection in the laser-induced choroidal neovascularization (L-CNV) mouse model that was used. The addition of the Fecm1Pas adds compelling supportive evidence to the FECH siRNA studies.

Referee #1 (Remarks):

The studies described in the manuscript entitled, "Ferrochelatase is a therapeutic target for ocular neovascularization", presents a detailed study in which the authors use a "forward chemical genetic approach" to find protein targets for cremastranone, a known anti-angiogenic natural product. The authors used photoaffinity chromatography to isolate cremastranone-bound proteins from tissue lysates of porcine brain. Ferrochelatase (FECH) was identified in this study. It is the terminal enzyme in heme biosynthesis and catalyzes the insertion of iron into protoporphyrin IX. Overexpression of FECH was found in human age-related macular degeneration (AMD) eyes. Inhibition of FECH with the oral anti-fungal drug, griseofulvin or the competitive inhibitor N-methyl protoporphyrin (NMPP) reduced endothelial cell migration and tube formation in several different types of retinal endothelial cell lines, as well as choroidal neovascularization and retinal endothelial sprouting in vitro. In vivo studies using a mouse laser-induced choroidal neovascularization model (L-CNV) showed loss of FECH expression by Fec siRNA, or a genetic loss of function Fecm1Pas mouse model is necessary for neovascularization. Inhibition of FECH by griseofulvin, either orally or by intravitreal injection, appears to result in similar reduction of neovascularization using the murine L-CNV model. Further studies indicated that depletion of FECH activity by Fec siRNA stabilizes hypoxia factor 1 α , which suppresses activation of VEGFR2 and by inference, neovascularization is inhibited. The authors concluded from these studies that FECH is a druggable mediator of angiogenesis whose inhibition could be used therapeutically to effectively block ocular neovascularization. In Figure 6, the authors attempt to tie in endothelial nitric oxide synthase (eNOS) into their model, claiming that the product of eNOS, nitric oxide, stabilizes HIF-1 α . The data in Figure 6 shows that depletion of FECH by Fec siRNA leads to a reduction of eNOS and decreased eNOS activity (A and B). In Figure 6C, the authors use hemin in a pull down assay of HRECs grown in either DMSO or 10 μ M NMPP after 1 week and show that eNOS levels are greater when cells are grown in the presence of the FECH competitive inhibitor, NMPP. Furthermore, they attributed the greater levels of eNOS to result from "an accumulation of apo-eNOS as demonstrated in a hemin pull down assay". The authors do not show that apo-eNOS increases under the conditions of the experiment.

Do the 2 graphs directly under Figure 6C belong with Fig 6C or are they a separate experiment? Fig 6A-B show eNOS activity and FECH siRNA inhibition are proportional in HRECs, i.e., when FECH levels decrease eNOS levels also decrease, but Fig 6C shows that growth of HRECs in the presence of NMPP increases eNOS. Nevertheless, in the 2 graphs below Fig 6C, despite FECH

depletion by Fech siRNA, exogenous addition of hemin, the enzymatic product of FECH, increases or rescues eNOS levels, but this couldn't possibly occur in vivo since heme is the enzymatic product of FECH. Given the contradictory results in Fig 6A-C, if eNOS levels are lower in the FECH siRNA depleted HRECs, that would mean there is LESS nitric oxide to stabilize HIF-1 α , not more, which weakly supports the HIF-1 α data in Figure 6D. How does decreased levels of nitric oxide tie in with the proposed model of FECH inhibition? The result in Fig 6C fits the logical reasoning that inhibiting or depleting FECH increases eNOS levels and nitric oxide so that HIF-1 α is effectively stabilized by nitric oxide and neovascularization is averted. The data in Fig 6A, B clearly supports decreased eNOS levels under the condition of FECH depletion by Fech siRNA, whereas the data in Fig 6C where FECH inhibition by NMPP shows increased eNOS levels. Perhaps the inhibition of the FECH gene or the inhibition of FECH modulates different regulatory/feedback pathways, transcriptionally or post-translationally.

The authors claim there is a slight additive effect when griseofulvin is injected with an anti-VEGF164 antibody (Fig EV5C), but there is no difference in CNV lesion volume in the last 3 bars of Figure 5C (dark red and 2 yellow bars). In fact, if there were an additive effect, the light yellow bar would be at least half the size of the 2nd blue bar. None of the last 3 bars are significantly different compared to injection of 50 μ m griseofulvin alone (blue bar). The only significant difference is the condition of injecting 0.2 μ m anti-VEGF164 vs. 0.2 μ m anti-VEGF164 + 50 μ m griseofulvin, which is a slight improvement but not additive (compared bright red and light yellow bars).

Specific Points

1. Fig 6C, need to show how levels of apo-eNOS were quantified, and if this correlates with the increased amount of eNOS in the hemin pull down assay.
2. On pp. 8, lines 176-178, the sentence "Despite the overall decrease in eNOS levels, we observed that there was an accumulation of apo-eNOS as demonstrated in a hemin pull down assay" needs to be rewritten. As written, this reviewer interpreted the pull down eNOS band in Fig 6C to be identified as apo-eNOS.
3. Fig EV2, panels in B and C look like the same cells but at a different magnification. Panels B should be labeled TUNEL and C should be labeled Caspase 3, respectively, to distinguish between the 2.
4. Fig EV2, Panel F, add to the legend in the top right (1 μ m Griseofulvin) to distinguish it from Panel D.
5. Fig EV4, Panel G, need to add a line to connect the data points.
6. FECH siRNA inhibits the Fech gene, which results in decreased FECH expression. On pp. 5, line 89 and pp. 7, line 144, change "FECH" to Fech and other places in the text and figures.
7. Anti-VEGF164 and Griseofulvin do not appear to be additive to decrease CNV lesion volume. On pp. 8, lines 165-168, need to state that synergy was not observed (or omit this altogether) and only a slight improvement was found when anti-VEGF164 and Griseofulvin used in combination.
8. On pp. 21, line 477, replace "wells were serum starved in..." with "medium was replaced with..."
9. In the Abstract, pp. 2, line 24, replace "...knockdown or mutation of Fech..." with "...siRNA knockdown of Fech or loss of function in the Fechm1Pas mouse model..."
10. In the Results section, pp. 4, lines 70 and 72, add the word "the" before "...the affinity reagent..."
11. In the Results section, pp. 7, line 141, invert word order of "...active site FECH..." to "FECH active site..."
12. In the Results section, pp. 7, line 145, replace, "...to have effects..." with "...to reduce neovascularization..."
13. In the Results section, pp. 7, line 146, place the last word "However, this..." with "Fortunately, the effective..."
14. In the Results section, pp. 7, line 151, replace "Nor did griseofulvin have..." with "Griseofulvin did not have..."
15. In the Results section, pp. 7, line 152, replace "...and HUVECs..." with "...or HUVECs..."
16. In the Results section, pp. 8, line 179, replace "As NO,..." with "Since NO,..." and "...causes stabilization of hypoxia..." with "stabilizes hypoxia..."
17. In the Results section, pp. 9, line 183, replace "As VEGF..." with "Since VEGF..." and delete the word "then" in "...we then monitored..."
18. In the Results section, pp. 9, lines 184, replace "...(activating) phosphorylation of the major VEGF receptor...." with "...active phosphorylated..."

19. In the Results section, pp. 9, lines 186 and 187, delete "...as growth factor..." and add "...(Fig 6E), 'whereas the' production..."

3rd Revision - authors' response

01 March 2017

Referee #1

The work described in the manuscript is interesting and presents ferrochelatase (FECH) as a possible new druggable target to prevent neovascularization in wet age-related macular degeneration. Oral delivery of the drug appears to be as effective, at least in the same order of magnitude, as intravitreal injection in the laser-induced choroidal neovascularization (L-CNV) mouse model that was used. The addition of the *Fechm1Pas* adds compelling supportive evidence to the FECH siRNA studies.

We thank the reviewer for this favourable assessment.

The studies described in the manuscript entitled, "Ferrochelatase is a therapeutic target for ocular neovascularization", presents a detailed study in which the authors use a "forward chemical genetic approach" to find protein targets for cremastranone, a known anti-angiogenic natural product. The authors used photoaffinity chromatography to isolate cremastranone-bound proteins from tissue lysates of porcine brain. Ferrochelatase (FECH) was identified in this study. It is the terminal enzyme in heme biosynthesis and catalyzes the insertion of iron into protoporphyrin IX. Overexpression of FECH was found in human age-related macular degeneration (AMD) eyes. Inhibition of FECH with the oral anti-fungal drug, griseofulvin or the competitive inhibitor N-methyl protoporphyrin (NMPP) reduced endothelial cell migration and tube formation in several different types of retinal endothelial cell lines, as well as choroidal neovascularization and retinal endothelial sprouting in vitro. In vivo studies using a mouse laser-induced choroidal neovascularization model (L-CNV) showed loss of FECH expression by *Fech* siRNA, or a genetic loss of function *Fechm1Pas* mouse model is necessary for neovascularization. Inhibition of FECH by griseofulvin, either orally or by intravitreal injection, appears to result in similar reduction of neovascularization using the murine L-CNV model. Further studies indicated that depletion of FECH activity by *Fech* siRNA stabilizes hypoxia factor 1 α , which suppresses activation of VEGFR2 and by inference, neovascularization is inhibited. The authors concluded from these studies that FECH is a druggable mediator of angiogenesis whose inhibition could be used therapeutically to effectively block ocular neovascularization.

Please note that, according to our data in Fig. 6D, *Fech* siRNA knockdown does *not* stabilize HIF-1 α , rather it leads to a decrease of this protein.

In Figure 6, the authors attempt to tie in endothelial nitric oxide synthase (eNOS) into their model, claiming that the product of eNOS, nitric oxide, stabilizes HIF-1 α . The data in Figure 6 shows that depletion of FECH by *Fech* siRNA leads to a reduction of eNOS and decreased eNOS activity (A and B). In Figure 6C, the authors use hemin in a pull down assay of HRECs grown in either DMSO or 10 μ M NMPP after 1 week and show that eNOS levels are greater when cells are grown in the presence of the FECH competitive inhibitor, NMPP. Furthermore, they attributed the greater levels of eNOS to result from "an accumulation of apo-eNOS as demonstrated in a hemin pull down assay". The authors do not show that apo-eNOS increases under the conditions of the experiment.

The results of the hemin pulldown are consistent with a reduction in total eNOS after FECH inhibition with NMPP (as seen on the input side of the blot), but what little eNOS remains is in the *apo* form, as only the *apo* (heme-free) form of hemoproteins will bind to hemin-agarose beads. We have now further clarified this point in Fig. 6, in the Fig. 6 legend, and in the text, pp. 8-9.

Do the 2 graphs directly under Figure 6C belong with Fig 6C or are they a separate experiment?

These two graphs belong with Fig 6E. We have adjusted the spacing of the figure to make this grouping clearer.

Fig 6A-B show eNOS activity and FECH siRNA inhibition are proportional in HRECs, i.e., when FECH levels decrease eNOS levels also decrease, but Fig 6C shows that growth of HRECs in the presence of NMPP increases eNOS.

As noted above and now more clearly explained in the text (pp. 8-9), the increase is only in the *apo* form of eNOS, but overall eNOS is depleted.

Nevertheless, in the 2 graphs below Fig 6C, despite FECH depletion by Fech siRNA, exogenous addition of hemin, the enzymatic product of FECH, increases or rescues eNOS levels, but this couldn't possibly occur in vivo since heme is the enzymatic product of FECH.

The addition of hemin shows that depletion of FECH can be partially rescued (normalized) by its enzymatic product in this in vitro context, confirming the relevance of FECH enzymatic activity to this process.

Given the contradictory results in Fig 6A-C, if eNOS levels are lower in the FECH siRNA depleted HRECs, that would mean there is LESS nitric oxide to stabilize HIF-1 α , not more, which weakly supports the HIF-1 α data in Figure 6D. How does decreased levels of nitric oxide tie in with the proposed model of FECH inhibition? The result in Fig 6C fits the logical reasoning that inhibiting or depleting FECH increases eNOS levels and nitric oxide so that HIF-1 α is effectively stabilized by nitric oxide and neovascularization is averted. The data in Fig 6A, B clearly supports decreased eNOS levels under the condition of FECH depletion by Fech siRNA, whereas the data in Fig 6C where FECH inhibition by NMPP shows increased eNOS levels. Perhaps the inhibition of the FECH gene or the inhibition of FECH modulates different regulatory/feedback pathways, transcriptionally or post-translationally.

Since *apo*-eNOS, as detected in Fig. 6C, is enzymatically inactive, all data in Figs. 6A-C are consistent: reduced FECH results in reduced eNOS protein levels, reduced eNOS hemylation, and reduced NOS activity. Since NO can stabilize HIF-1 α (Sandau et al., *Blood*, 2001), reduced eNOS might be expected to be associated with reduced HIF-1 α , which is what we observed on FECH knockdown (Fig. 6D).

The authors claim there is a slight additive effect when griseofulvin is injected with an anti-VEGF164 antibody (Fig EV5C), but there is no difference in CNV lesion volume in the last 3 bars of Figure 5C (dark red and 2 yellow bars). In fact, if there were an additive effect, the light yellow bar would be at least half the size of the 2nd blue bar. None of the last 3 bars are significantly different compared to injection of 50 μ M griseofulvin alone (blue bar). The only significant difference is the condition of injecting 0.2 μ M anti-VEGF164 vs. 0.2 μ M anti-VEGF164 + 50 μ M griseofulvin, which is a slight improvement but not additive (compared bright red and light yellow bars).

As the reviewer notes, there is a slight improvement in CNV with the combination of griseofulvin and 0.2 ng anti-VEGF versus each agent alone (the yellow bar is lower than either the blue or red bars). We would argue that this justifies our assertion of a "modest additive effect", but as we now note in the text (p. 8), this should be explored further in future.

Specific Points

1. Fig 6C, need to show how levels of apo-eNOS were quantified, and if this correlates with the increased amount of eNOS in the hemin pull down assay.

Please see above for clarification of this point.

2. On pp. 8, lines 176-178, the sentence "Despite the overall decrease in eNOS levels, we observed that there was an accumulation of apo-eNOS as demonstrated in a hemin pull down assay" needs to be rewritten. As written, this reviewer interpreted the pull down eNOS band in Fig 6C to be identified as apo-eNOS.

The reviewer is correct in this interpretation of the text as written; please see above for details.

3. Fig EV2, panels in B and C look like the same cells but at a different magnification. Panels B should be labeled TUNEL and C should be labeled Caspase 3, respectively, to distinguish between the 2.

We have double-checked and confirmed that these panels present the correct data. We have added the suggested labels.

4. Fig EV2, Panel F, add to the legend in the top right (1 μ M Griseofulvin) to distinguish it from Panel D.

We have added the suggested label to this and the other panels.

5. Fig EV4, Panel G, need to add a line to connect the data points.

We have added the suggested line, although a true dose-response curve could not be fitted to this particular data set.

6. FECH siRNA inhibits the *Fech* gene, which results in decreased FECH expression. On pp. 5, line 89 and pp. 7, line 144, change "FECH" to *Fech* and other places in the text and figures.

We have checked all usages in the text and confirm that our usage is consistent with HGNC and MGI recommendations: italicized *FECH* for the human DNA and mRNA, italicized *Fech* for the mouse DNA and mRNA, and Roman FECH for the protein of both species.

7. Anti-VEGF164 and Griseofulvin do not appear to be additive to decrease CNV lesion volume. On pp. 8, lines 165-168, need to state that synergy was not observed (or omit this altogether) and only a slight improvement was found when anti-VEGF164 and Griseofulvin used in combination.

Please see above for discussion of this point.

8. On pp. 21, line 477, replace "wells were serum starved in..." with "medium was replaced with..."

Revised as suggested.

9. In the Abstract, pp. 2, line 24, replace "...knockdown or mutation of *Fech*..." with "...siRNA knockdown of *Fech* or loss of function in the *Fech*^{m1Pas} mouse model..."

Revised as suggested.

10. In the Results section, pp. 4, lines 70 and 72, add the word "the" before "...the affinity reagent..."

Revised as suggested.

11. In the Results section, pp. 7, line 141, invert word order of "...active site FECH..." to "FECH active site..."

Revised as suggested.

12. In the Results section, pp. 7, line 145, replace "...to have effects..." with "...to reduce neovascularization..."

We have clarified this text with alternative wording as we are not discussing neovascularization here.

13. In the Results section, pp. 7, line 146, place the last word "However, this..." with "Fortunately, the effective..."

We have left this unchanged to avoid the bias implied by “fortunately”.

14. In the Results section, pp. 7, line 151, replace "Nor did griseofulvin have..." with "Griseofulvin did not have..."

Revised as suggested.

15. In the Results section, pp. 7, line 152, replace "...and HUVECs..." with "...or HUVECs..."

Revised as suggested.

16. In the Results section, pp. 8, line 179, replace "As NO,..." with "Since NO,.." and "...causes stabilization of hypoxia..." with "stabilizes hypoxia..."

Revised as suggested.

17. In the Results section, pp. 9, line 183, replace "As VEGF..." with "Since VEGF..." and delete the word "then" in ...we then monitored..."

Revised as suggested.

18. In the Results section, pp. 9, lines 184, replace "...(activating) phosphorylation of the major VEGF receptor...." with "...active phosphorylated..."

We have left this unchanged as the suggested change would require complete reworking of this sentence.

19. In the Results section, pp. 9, lines 186 and 187, delete "...as growth factor..." and add "...(Fig 6E), 'whereas the' production..."

Revised with similar wording to the suggestion.

We here upload the revised manuscript, six Figures, five Expanded View figures, and the Appendix containing two Supplementary Figures. We also re-upload the Source Data files as we note that some formatting issues arose in the previous conversion to PDFs, and we have made a minor edit (addition of another loading control blot) to Fig. 6E and the corresponding Source Data. Thank you again for accepting our manuscript.

Corresponding Author Name: Corson
Journal Submitted to: EMBO Molecular Medicine
Manuscript Number: EMM-2016-06561